

# The evolution of Arctic permafrost over the last three centuries

Moritz Langer[1,2], Jan Nitzbon[1,3], Brian Groenke[1,4], Lisa-Marie Assmann[1], Thomas Schneider von Deimling[1,2], Simone Maria Stuenzi[1], and Sebastian Westermann[5]

[1]Permafrost Research Section, Alfred Wegener Institute Helmholtz Centre for Polar and Marine Research, Potsdam, Germany
[2]Geography Department, Humboldt-Universität zu Berlin, Berlin, Germany
[3]Paleoclimate Dynamics Section, Alfred Wegener Institute Helmholtz Centre for Polar and Marine Research, Bremerhaven, Germany
[4]Department of Electrical Engineering and Computer Science, Technical University of Berlin, Berlin, Germany
[5]Department of Geosciences, University of Oslo, Oslo, Norway

**Correspondence:** M. Langer (moritz.langer@awi.de)

**Abstract.** Understanding the future evolution of permafrost requires a better understanding of its climatological past. This requires permafrost models to efficiently simulate the thermal dynamics of permafrost over the past centuries to millennia, taking into account highly uncertain soil and snow properties. In this study, we present a computationally efficient numerical permafrost model which satisfactorily reproduces the current thermal state of permafrost in the Arctic and its recent trend

over the last decade. Also, the active layer dynamics and its trend is realistically captured. The performed simulations provide insights into the evolution of permafrost since the 18th century and show that permafrost on the North American continent is subject to early degradation, while permafrost on the Eurasian continent is relatively stable over the investigated 300-year period. Permafrost warming since industrialization has occurred primarily in three "hotspot" regions in northeastern Canada, northern Alaska, and, to a lesser extent, western Siberia. The extent of near-surface permafrost has changed substantially since

the 18th century. In particular, loss of continuous permafrost has accelerated from low ($-0.10 \times 10^5 \, \mathrm{km^2 \, dec^{-1}}$) to moderate ($-0.77 \times 10^5 \, \mathrm{km^2 \, dec^{-1}}$) rates for the 18th and 19th centuries, respectively. In the 20th century, the loss rate nearly doubled ($-1.36 \times 10^5 \, \mathrm{km^2 \, dec^{-1}}$), with the highest near-surface permafrost losses occurring in the last 50 years. Our simulations further indicate that climate disturbances due to large volcanic eruptions in the Northern Hemisphere, can only counteract near-surface permafrost loss for a relatively short period of a few decades. Despite some limitations, the presented model shows

great potential for further investigation of the climatological past of permafrost, especially in conjunction with paleoclimate modeling.

## 1   Introduction

With an area of about 12 to 17 million square kilometers (Gruber, 2012; Chadburn et al., 2017; Obu et al., 2019), permafrost is the largest, non-seasonal component of the Earth's cryosphere. Permafrost is a characteristic factor of Arctic and subarctic

ecosystems and determines a variety of fundamental hydrological and biogeochemical processes (Walvoord and Kurylyk, 2016; Turetsky et al., 2020). The occurrence and thermal state of permafrost is determined by long-term local climate history (French and Millar, 2014). In particular, the thermal state of the deeper soil layers (i.e. in depths of tens to hundreds of meters) must



be considered a legacy of a past climate (e.g. Kneier et al., 2018). The oldest known permafrost has survived 650,000 years
of glacial-interglacial climate cycles, persisting through substantial climatic changes (Murton et al., 2022). Thus, present-day
permafrost formed mainly over tens to hundreds of thousands of years under colder climatic conditions which, in combination
with sedimentation, led to accumulation of ground ice and organic carbon at increasing soil depths (Kanevskiy et al., 2017).
Although it is assumed that most of the ground ice and organic material occurs near the surface (<3 m) (Hugelius et al., 2014)
deeper reservoirs of organic carbon and ground ice do exist and control the long term thaw sensitivity of permafrost landscapes
(West and Plug, 2008). For the Alaskan coastal plain, Péwé (1979) estimated that pore and segregated ground ice comprise
41% by volume of the soil at depths between 3 and 10 m. Ice wedges extend even deeper into the ground, with very deep and
massive deposits of ground ice and carbon found in the Yedoma deposits of Siberia and Alaska, reaching depths over 50 m
(Kanevskiy et al., 2011; Strauss et al., 2021). The presence of ground ice and carbon at greater soil depths demands a better
understanding of the thermal state of deep permafrost and its sensitivity towards climatic changes.

Reconstruction of the climatically induced thermal state of permafrost at depths of about 50 m requires simulation times
of centuries to millennia to achieve a dynamic equilibrium that can be considered largely unaffected by initial conditions
(e.g. Ross et al., 2021). This requires computationally efficient models, in particular if large regions represented by many
grid cells are to be evaluated. In addition, it is important that such models can operate with limited and highly uncertain
information about thermal and hydrological ground properties. In particular, ground water and ground ice contents are highly
uncertain, but strongly affect heat uptake and storage in the ground. Such uncertainties can be addressed with probabilistic
approaches such as parameter ensemble simulations (e.g. Schneider von Deimling et al., 2006; Murphy et al., 2007), which
allow simulations to be evaluated with the consideration of plausible parameter ranges. However, simulations of a large number
of ensemble members require efficient computations and prefer a small amount of required input data. The same is true for other
probabilistic model applications such as data assimilation and hybrid modeling (e.g. Madadgar and Moradkhani, 2014; Kraft
et al., 2022). Furthermore, climate reconstructions usually provide only large-scale air temperature and precipitation data, while
other near-surface climate variables such as wind speed, air pressure, specific humidity, and radiation are difficult to obtain in
particular at high temporal resolution. This severely limits the options for model forcing and hence process representation (e.g.
due to unclosed surface energy and water balances). Simulations targeting the long-term evolution of deep permafrost therefore
impose certain requirements and constraints on the model used.

Numerous permafrost models are available ranging from analytical steady-state solutions (e.g. Gruber, 2012; Obu et al.,
2019) to very sophisticated thermo-hydrological numerical models (e.g. Kurylyk and Watanabe, 2013; Karra et al., 2014;
Atchley et al., 2015; Westermann et al., 2022). The latter usually require an immense computational effort for experiments
spanning hundreds of years and are, therefore, typically used for local to regional process studies covering periods between
years and decades. In contrast, Pan-Arctic permafrost simulations, such as those performed with Earth System Models (ESMs),
typically comprise several hundreds of years for historical and future climate projections. Many land surface schemes from
ESMs cannot capture the long-term thermal evolution of deep permafrost since their representation of the ground is limited to
the upper meters (Hermoso de Mendoza et al., 2020; Steinert et al., 2021). To date, there are only few permafrost modeling ap-
proaches that focus on paleoclimatic periods (e.g. Crichton et al., 2014; Willeit and Ganopolski, 2015; Schneider von Deimling



et al., 2018; Kitover et al., 2019; Saito et al., 2021). The ability of such models to perform simulations over millennia comes at the price of a very limited representation of processes and low spatial and temporal resolution.

Here, we present and evaluate a computationally efficient numerical permafrost model designed to provide insights into the evolution of the thermal state of permafrost and active layer thickness over many centuries for the Arctic permafrost region. Our approach accounts for uncertainties in soil parameters such as water and ice content, and uncertainties in snowpack properties through parameter ensemble simulations. With this approach, we aim to bridge the gap between very sophisticated permafrost process models and reduced schemes used for paleoclimatic simulations. Based on observations, we investigate the ability

of the new model to reproduce the current thermal state of permafrost as determined by the climatic evolution over the last centuries.The required model parameters are greatly reduced compared to permarost process models and are based entirely on pan-Arctic or global datasets. Model forcing is limited to daily mean surface temperature, precipitation, and geothermal heat flux. Specifically, we evaluate the applicability and performance of the model to represent the current thermal state of permafrost and its temperature trend in recent decades based on long-term temperature records from boreholes. In addition, we

use observations of the thickness of the active layer to evaluate the model's ability to reproduce annual freeze-thaw dynamics. In addition, we apply the model to trace the evolution of Arctic permafrost over the past three centuries.

## 2   Methods

CryoGridLite largely adopts schemes and parameterizations previously implemented and tested in the context of regional and local permafrost modeling using CryoGrid2 (Westermann et al., 2011; Langer et al., 2013; Westermann et al., 2017) and

CryoGrid3 (Westermann et al., 2016). CryoGridLite consists of (i) a soil module that calculates conductive heat transfer with phase change and (ii) a dynamic snow scheme to represent the insulating effect of seasonal snow cover. Furthermore, some data prepossessing is demanded in order to derive gridded soil stratigraphies and snow parameters by synthesizing different global datasets, and to generate offline model forcings based on global reanalysis data and paleoclimate simulations.

### 2.1   Ground heat transfer and phase change

In contrast to its predecessors, CryoGrid2 and CryoGrid3, CryoGridLite solves the nonlinear heat equation with phase change in terms of enthalpy density (volumetric enthalpy) instead of temperature:

$$\frac{\partial H_v(T)}{\partial t} - \nabla_z [K(T) \nabla_z T] = 0 \qquad (1)$$

where $H_v$ [J m$^{-3}$] is the volumetric enthalpy as an invertible function of temperature $T$ [K], $z$ [m] is depth along the vertical axis, $t$ [s] is time, and $K(T)$ [W m$^{-1}$ K$^{-1}$] is the temperature-dependent thermal conductivity. Equation (1) can be solved

using the iterative, backward Euler scheme given by Swaminathan and Voller (1992):

$$\alpha_j^i T_{j-1}^{i+1} - \beta_j^i T_j^{i+1} + \gamma_i^i T_{j+1}^{i+1} + b_j^i - \frac{H_{v j}^{i+1} - H_{v j}^0}{\Delta t} = 0, \qquad (2)$$



with

$$\frac{{H_\mathrm{v}}_j^{i+1} - {H_\mathrm{v}}_j^{0}}{\Delta t} = \left(\frac{\mathrm{d}H_\mathrm{v}}{\mathrm{d}T}\right)_j^i \frac{T_j^{i+1} - T_j^i}{\Delta t} + \frac{{H_\mathrm{v}}_j^{i} - {H_\mathrm{v}}_j^{0}}{\Delta t}, \tag{3}$$

where $\Delta t$ [s] is a constant time step, $T$ [K] is the temperature, $\alpha$, $\beta$, and $\gamma$ [$\mathrm{J\,m^{-3}\,K^{-1}\,s^{-1}}$] are pre-factors determined by the
temporal invariant grid cell spacing and variable thermal conductivities, and $b$ [$\mathrm{J\,m^{-3}\,s^{-1}}$] is an optional energy source term
which can be used to apply an external forcing. The index $j$ marks the grid cell number increasing with depth, and the index $i$
indicates the iteration step. The iteration step $i = 0$ indicates the state of the previous time step $(t - 1)$ which is updated to the
current time step $(t)$ after reaching convergence.

At each time step, iteration continues until the temperature profile matches the inverse enthalpy profile controlled by a
tolerance parameter which we set to $1 \times 10^{-3}$ K. The release and uptake of latent heat during phase change is accounted for by
the derivative of the enthalpy function, $\frac{dH_\mathrm{v}}{dT}$. We choose the form of the enthalpy function to follow the freezing characteristic
of pure water (i.e. "free" water; for details, see Appendix A). This has the benefit of providing a readily available inverse
function (Eq. A3) which is necessary for the iterative scheme, at the cost of neglecting complex interactions of the freezing
characteristic with soil composition, structure, and chemistry (Koopmans and Miller, 1966). For each grid cell the ground
thermal properties such as thermal conductivities and heat capacities are calculated based on the actual ground composition
defined by the volumetric contents of organic, mineral, water, and ice. The upper boundary condition is defined by an external
surface temperature, while the lower boundary condition is defined by a locally constant geothermal heat flux (Davies, 2013).
In rare cases where convergence can not be reached within a maximum of 500 iterations, the maximum temperature deviation
is printed out as a warning before the next time step is calculated. It is, however, important to note that such a temperature
deviation only indicates that the current temperature profile is not consistent with the enthalpy profile. This deviation does not
affect energy conservation over the total profile but indicates that the distribution of energy within the profile is not exact.

Numerical performance and stability of the applied implicit scheme is evaluated against an equivalent model configuration
solved using several standard numerical integration schemes: Crank-Nicolson (2nd order, diagonally implicit), Radau IIA (5th
order, fully implicit), and stabilized Runge-Kutta (4th order, explicit), all of which require sub-daily time steps during the
thawing season in order to maintain stability. The test simulations under constant freezing and thawing conditions reveal root
mean square errors below $0.01$ K. For more details about the numerics and performance of the applied solving scheme we
refer to Swaminathan and Voller (1992). We note that the implicit time stepping scheme employed in this work bears some
resemblance to a recently proposed method (Tubini et al., 2021) which applies the Newton-Casuli-Zanolli algorithm (Casuli
and Zanolli, 2010) to solve the nonlinear heat equation in enthalpy form. The primary advantage of the applied scheme is
that it is found to be generally stable using daily time steps while still being energy-conserving. It is furthermore capable of
representing the latent heat effect via a simplified freezing characteristic without requiring the computationally demanding
task of explicitly tracking the position of freezing fronts. We note that the implicit solver used is strictly valid only for phase
change within a homogeneous material. However, we point out that any uncertainties due to natural heterogeneities at the
surface and in the subsurface on the spatial scales considered are likely to be larger than the errors introduced by a generalized




representation of the heat transfer process. Conservation of energy is the primary concern for long-term simulation between centuries and millennia.

### 2.1.1 Snowpack representation

The snowpack is an important factor controlling the thermal state of permafrost and induces large uncertainties in permafrost modeling in general (Langer et al., 2013; Gouttevin et al., 2018; Jan and Painter, 2020). The insulating effect of the snowpack

must therefore be carefully represented, although very coarse spatial model resolutions and reduced forcing data limit the ability to simulate complex processes determining the snow properties. CryoGridLite simulates the insulative effect of the snowpack using a bulk approach based on daily snowfall rates and snow properties that are defined by climate regions following Sturm et al. (2010). Snow accumulates on top of the ground domain according to snowfall. For this purpose, empty ghost cells on top of the ground domain are populated by snow with an initial snow density according to snow depth. The snow depth is updated

at the beginning of each time step as

$$h_{\text{snow}}^t = \left( h_{\text{snow}}^{t-1} - \Delta h_{\text{melt}}^{t-1} + P_{\text{snow}}^t \, \Delta t \, \frac{\rho_{\text{water}}}{\rho_{\text{snow,min}}} \right) \frac{\overline{\rho}_{\text{snow}}^t}{\rho_{\text{snow,bulk}}^t}, \tag{4}$$

where $h_{\text{snow}}$ [m] is the snow depth, $\Delta h_{\text{melt}}$ [m] is the change in snow depth due to melting, $P_{\text{snow}}$ [m s$^{-1}$] is the snowfall rate, $\rho_{\text{water}}$ [kg m$^{-3}$] is the density water (set to $1000 \, \text{kg m}^{-3}$), $\rho_{\text{snow,min}}$ is the initial snow density, $\overline{\rho}_{\text{snow}}$ is the average density of the actual snowpack, and $\rho_{\text{snow,bulk}}$ is the bulk snow density predicted according to Sturm et al. (1995) as

$$\rho_{\text{snow,bulk}}^t = (\rho_{\text{snow,max}} - \rho_{\text{snow,min}})(1 - e^{-k_1 \, h_{\text{snow}}^t - k_2 \, d^t}) + \rho_{\text{snow,min}}, \tag{5}$$

where $\rho_{\text{snow,max}}$ is the maximum snow density, $k_1$, and $k_2$ are empirical snow densification factors, and $d$ indicates the day count of the snow season. All snow parameters are set according to values introduced by Sturm et al. (2010) defined by a global map of snow-climate zones. The snow density of each grid cell is scaled so that the average density of the actual snowpack matches the predicted bulk snow density. This procedure generates a layered snowpack over time with highest densities for old

snow layers at the bottom of the snowpack. Heat transfer and phase change in the snowpack are calculated simultaneously with the ground using the implicit scheme described above. The ablation of the snow cover is calculated with a positive degree-day approach very similar to Tsai and Ruan (2018). The snow scheme further accounts for melt-water infiltration and refreezing similar to Westermann et al. (2011) using a diagnostic bucket scheme. The bucket scheme assumes instantaneous melt-water routing if a maximum water holding capacity of snow is exceeded. Water that exceeds a maximum value of water saturation

is routed away so that ponding of meltwater is precluded. A maximum snow depth is defined by a threshold value ($h_{\text{snow,max}}$) which can be used to emulate wind-induced snow depth limitation as observed, for example, on ice-covered lakes (Sturm and Liston, 2003) or other wind-exposed parts of the landscape. The snow scheme is restricted to simulating seasonal snow by setting snow depth to zero in August of each simulation year. The build-up of multi-annual snowpacks is thus precluded.





## 2.2 Ground stratigraphies

### 2.2.1 Subsurface layers and vertical model grid

The ground stratigraphy is represented by six ground layers characterized by different volume fractions of soil constituents (Fig. 1). Thereby two constant layer boundaries ($z_{\mathrm{SOC30}}$, $z_{\mathrm{SOC300}}$) are defined according to the soil layers specified in the Northern Circumpolar Soil Carbon Database v2 (NCSDv2) (Hugelius et al., 2013). Three spatially variable soil layer boundaries marking the root zone ($z_{\mathrm{R}}$), vadose zone ($z_{\mathrm{V}}$), and saturated zone ($z_{\mathrm{S}}$) are defined by soil thickness data from the Gridded Global Data Set of Soil Thickness (Pelletier et al., 2016) and specifications of the water table (see Sect. 2.3).

Regardless of the soil layers described above, the vertical model domain used for numerical integration is discretized into about 400 cells from the surface at $z = 0\,\mathrm{m}$ to a maximum depth of $550\,\mathrm{m}$. The spacing of the grid cells increases with depth in an approximately logarithmic fashion, allowing a very high vertical resolution of $0.01\,\mathrm{m}$ near the surface, while the lowest grid cells are $100\,\mathrm{m}$ thick. To represent the temporal accumulation of snow above the ground surface, the vertical grid contains 200 additional ghost cells that allow the snow layer to be simulated at a vertical resolution of $0.01\,\mathrm{m}$.

### 2.2.2 Soil stratigraphy

The vertical distribution of the ground constituents is specified based on parameterizations used for the SURFEX land surface and ocean scheme (Masson et al., 2013). The SURFEX parameterization approximates a profile of the organic soil fraction as

$$f_{\mathrm{o}}^{i} = \frac{\rho_{\mathrm{SOC}}^{i}}{(1 - \phi_{\mathrm{o}}^{i})\,\rho_{\mathrm{om}}}, \tag{6}$$

and a profile of the mineral soil fraction as

$$f_{\mathrm{m}}^{i} = 1 - f_{\mathrm{o}}^{i}, \tag{7}$$

where the index $i$ denotes here the ground layers, $\rho_{\mathrm{SOC}}$ is the soil organic carbon density, $\rho_{\mathrm{om}}$ is the pure organic matter density set to $1300\,\mathrm{kg\,m^{-3}}$, and $\phi_{\mathrm{o}}$ $[\mathrm{m^3\,m^{-3}}]$ is the volumetric porosity of organic soil decreasing from 0.93 to 0.84 with depth following a power function (see supplement in Masson et al., 2013). Despite the fact that the model does not include a water balance scheme, we use hydrological soil parameters (wilting point ($\theta_{\mathrm{wp}}$), field capacity ($\theta_{\mathrm{fc}}$)) to specify the vertical water and ice distribution within the ground (see Sect. 2.3). In this context, the power function used to scale the organic soil porosity with depth is also applied to scale the organic soil wilting point ($\theta_{\mathrm{wp,o}}$) between 0.07 and 0.22 and the organic soil field capacity ($\theta_{\mathrm{fc,o}}$) between 0.37 and 0.72. The required data on soil organic carbon density are taken from the NCSDv2 (Hugelius et al., 2013).



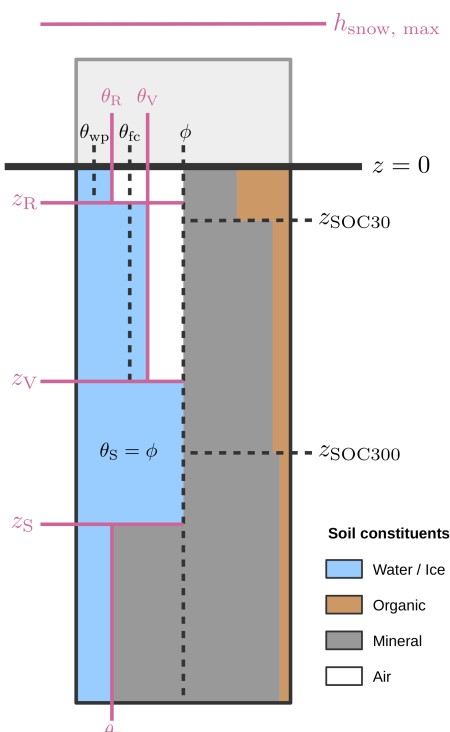

**Figure 1.** Schematic illustration of the applied ground stratigraphy determined by the volumetric ground constitutes, derived based on various global datasets. Layer boundaries that control the vertical distribution of groundwater, ground ice, and snow (marked in red) are randomly varied to generate a parameter ensemble. This way a wide range of plausible hydrological conditions is represented by the performed simulations. See Table 1 for the meaning of the symbols.

175    The mineral soil porosity ($\phi_{\mathrm{m}}$), the wilting point ($\theta_{\mathrm{wp,m}}$), and the field capacity ($\theta_{\mathrm{fc,m}}$) were approximated based on the SURFEX parameterizations as

$$\phi_{\mathrm{m}} = 0.49 - 0.11 \, f_{\mathrm{sand}}, \tag{8}$$

$$\theta_{\mathrm{wp,m}} = 0.37 \sqrt{f_{\mathrm{clay}}}, \tag{9}$$

$$\theta_{\mathrm{fc,m}} = 0.45 \left( f_{\mathrm{clay}} \right)^{0.3496}, \tag{10}$$

180    with the fractions of sand and clay obtained from the Open-ECOCLIMAP database (Masson et al., 2003; Faroux et al., 2013). The combined soil porosities ($\phi$), field capacities ($\theta_{\mathrm{fc}}$), wilting points ($\theta_{\mathrm{wp}}$), volumetric mineral contents ($\theta_{\mathrm{m}}$), and the volu-





metric organic contents ($\theta_o$) are then calculated as weighted means of the organic and mineral soil fractions:

$$\phi^i = f_o^i \phi_o^i + f_m^i \phi_m, \tag{11}$$

$$\theta_{fc}^i = f_o^i \theta_{fc,o}^i + f_m^i \theta_{fc,m}, \tag{12}$$

$$\theta_{wp}^i = f_o^i \theta_{wp,o}^i + f_m^i \theta_{wp,m}, \tag{13}$$

$$\theta_o^i = f_o^i (1 - \phi_o^i), \tag{14}$$

$$\theta_m^i = f_m^i (1 - \phi_m). \tag{15}$$

The water/ice contents ($\theta_w$) of the different soil layers are varied during parameter ensemble simulations (see Section 2.3) within constraints provided by the hydrological parameters above (see Table 1).

### 2.2.3 Ground thermal properties

The thermal properties of the subsurface layers (thermal conductivity and volumetric heat capacity) are parameterized based on their composition following Westermann et al. (2013). Both volumetric heat capacity and the thermal conductivity are functions of the volumetric fractions of the ground constituents. The volumetric heat capacity $C_v$ is calculated as weighted arithmetic mean as

$$C_v = \sum_{n=1}^{N} \theta_n c_n, \tag{16}$$

where $N$ is the total number of soil constitutes in the mixture, $\theta_n$ is the volumetric content of the $n$-th soil constituent, and the volumetric heat capacity of each soil constituent $c_n$ is set according to values provided in Tab. A1.

The thermal conductivity is approximated using a quadratic parallel model (Cosenza et al., 2003) as

$$K = \left( \sum_{n=1}^{N} \theta_n \sqrt{k_n} \right)^2, \tag{17}$$

the thermal conductivity of each soil constituent is set to values provided in Tab. A1. Note that the thermal conductivity model can be easily replaced by any other model used to approximate multiphase thermal conductivities of ground.

### 2.3 Parameter ensemble simulations

Since CryoGridLite does not have a dedicated hydrology scheme and because of the low confidence in the available datasets and parameterizations on ground water and ground ice contents at high latitudes, we apply parameter ensemble simulations to represent a wide range of hydrological conditions. Here, the relevant factors controlling the vertical water and ice distribution in the ground profile are randomly varied within realistic ranges. In this way, an ensemble of $n = 50$ randomly sampled independent parameter settings is simulated for each model grid cell.

The default ground water and ground ice content in the subsurface is parameterized such that it reflects average hydrological conditions. For this we distinguish four hydrologically different zones in the subsurface: the root zone (R), the vadose zone (V),



**Table 1.** Overview of the parameters which were varied in the parameter ensemble simulations. The parameters were independently drawn from a uniform distribution between the respective minimum and maximum values. Here, $Z_{\mathrm{mean}}$ and $Z_{\mathrm{max}}$ refer to the mean and max soil thickness respectively.

| Parameter | Symbol | Spin-up value | Ens. min. value | Ens. max. value |
|---|---|---|---|---|
| Maximum snow height | $h_{\mathrm{snow,max}}$ | $2\,\mathrm{m}$ | $0.1\,\mathrm{m}$ | $2\,\mathrm{m}$ |
| Root zone depth | $z_{\mathrm{R}}$ | $0.04\,\mathrm{m}$ | $0\,\mathrm{m}$ | $0.2\,\mathrm{m}$ |
| Root zone water content | $\theta_{\mathrm{R}}$ | $\frac{\theta_{\mathrm{wp}}+\theta_{\mathrm{fc}}}{2}$ | $\theta_{\mathrm{wp}}$ | $\theta_{\mathrm{fc}}$ |
| Vadose zone depth | $z_{\mathrm{V}}$ | $Z_{\mathrm{mean}}$ | $z_{\mathrm{R}}^{*}$ | $z_{\mathrm{S}}^{*}$ |
| Vadose zone water content | $\theta_{\mathrm{V}}$ | $\frac{\theta_{\mathrm{fc}}+\phi}{2}$ | $\theta_{\mathrm{fc}}$ | $\phi$ |
| Saturated zone depth (below is bedrock) | $z_{\mathrm{S}}$ | —** | $Z_{\mathrm{mean}}$ | $Z_{\mathrm{max}}$ |
| Bedrock zone water content*** | $\theta_{\mathrm{B}}$ | $\phi$ | $0$ | $\phi$ |

*Needs to be sampled before $z_V$.

**Extends down to the end of the model domain, i.e. no bedrock with reduced ice content.

***Note that the remaining pore space is filled with mineral sediment, i.e. effectively the pore space in the bedrock zone is reduced and saturated with ice.

the saturated zone (S) and the bedrock zone (B) (see Fig. 1). By default (i.e., during model spin-up), the water content is set halfway between the wilting point and the field capacity of the respective soil layer(s) in the root zone, and halfway between field capacity and porosity in the vadose zone. In the saturated zone, the ground layers are always completely saturated. In addition, we assume a saturated bedrock zone which starts at a depth derived from the Gridded Global Data Set of Soil Thickness (Pelletier et al., 2016). Below this depth ($z_S$), the water and ice content is reduced in favor of an increased mineral

content. The default parameters defining the ground water and ice contents are provided in Table 1.

To induce variation in the vertical ground water and ground ice distribution, the root zone depth ($z_R$), vadose zone depth ($z_V$) and saturated zone depth ($z_S$) are drawn from a uniform distribution between the minimum and maximum values given in Table 1. The depth range for the root zone is set based on field experience from the authors; the depth of the saturated zone which corresponds to the overall soil thickness above the bedrock is varied between the mean and the maximum soil thickness

estimates contained in the respective $1°$ by $1°$ grid cell from the dataset of Pelletier et al. (2016). The border between the vadose and saturated zone is set to a random value between $z_R$ and $z_S$ after these were drawn.

In addition to a variation of the ground ice contents through the parameters described above, we also vary the maximum snow height, a threshold value which limits the height of the snowpack. While the variation of ground ice contents corresponds mainly to a variation of the (potential) latent heat content of the ground, the maximum snow height excerpts strong control

on the amount of sensible heat stored in the subsurface. This parameter variation in the model ensemble accounts for heterogeneities in the micro- and meso-scale topography, which result in highly variable snowpack heights in reality (e.g. Zweigel et al., 2021).



## 2.4 Climate forcing

The model uses an external climate forcing of daily averages of surface temperature and precipitation. This greatly simplifies
its application to very long time series spanning centuries to millennia. We consider daily mean near-surface air temperatures
as appropriate first-order estimates of surface temperatures. The snowfall is estimated from total precipitation that falls when
air temperatures are below $0\,°C$. The applied climate forcing consists of a synthesized time series of daily mean surface
air temperatures and total precipitation combining paleoclimate simulations (500 A.D. - 1979) and reanalysis data (1979 -
2019). We use paleoclimate simulations from the Commonwealth Scientific and Industrial Research Organisation (CSIRO)
based on the Climate System Model Mk3Lv.1.2 from which we selected the ensemble including Orbital, Greenhouse gas,
Solar, and Volcanic (OGSV) forcing due to its improved capability to reproduce climate reconstructions for the northern
hemisphere and the inclusion of climate events such as volcanic eruptions (Phipps et al., 2013). From the OGSV ensemble
we arbitrarily select an ensemble member (which differ only in initial conditions) to generate the forcing for our simulations.
For the reanalysis period we applied ERA-Interim data (Dee et al., 2011). The paleo simulations were harmonized with the
ERA-Interim baseline data by applying decadal-mean monthly anomalies of temperature and precipitation to the first decade
(1980-1990) of the reanalysis data. The forcing at the lower boundary of the ground domain in $550\,m$ depth is determined
by a local geothermal heat flux according to the Global Map of Solid Earth Surface Heat Flow (Davies, 2013). The spatial
resolution of the synthesized climate forcing data was set to $1\,°C$ demanding spatial harmonization of the different forcing
data. We applied spatial averaging for the ERA-Interim data (with nominal resolution of $\approx 80\,km$), and nearest neighbor
interpolation for CSIRO-Mk3Lv.1.2 (with nominal resolution of $5.68°$ for longitude and $3.28°$ for latitude) as well as for the
geothermal heat flux map (with a nominal resolution of $2°$). An overview of the input data used for model forcing and model
parameterization is provided in Table 2.

## 3 Results and discussion

### 3.1 Model evaluation

### 3.1.1 Ground temperatures

The ability of the model to reproduce the ground thermal regime and temperature trends in the Arctic permafrost region is
evaluated using borehole temperature measurements from the Global Terrestrial Network for Permafrost (GTN-P) (Biskaborn
et al., 2019). The dataset comprises $n = 82$ boreholes within our model domain for which it provides observations of mean
annual ground temperatures (MAGT) for the period from 2007 to 2016.
Direct comparison between observed and modeled ground temperatures shows that at most sites ($65.9\,\%$) observed MAGT
can be reproduced with deviations of up to $\pm 2\,K$ (Fig. 2a), in particular when the model range resulting from the parameter
ensemble is considered. However, we note this relatively good agreement between observations and model is partly due to a
number of boreholes showing temperatures at or near $0\,°C$ (Fig. 2b). Because of the phase change of the ground ice and its



**Table 2.** Overview of datasets used to force and parameterize CryoGridLite.

| Input parameter | Dataset | Source/Reference | Comments |
| --- | --- | --- | --- |
| Meteorological forcing | ERA-Interim | Dee et al. (2011) | baseline forcing 1979-2019, downsampled to 1°x1° by spatial averaging |
| Paleo climate anomalies | CSIRO - Mk3Lv.1.2 | Phipps et al. (2013) | 500 A.D. - 1979, upsampled to 1°x1° by nearest neighbor interpolation |
| Geothermal heat flux | Global Map of Solid Earth Surface Heat Flow | Davies (2013) | lower boundary conditions |
| Volumetric ground composition, porosity, field capacity | SURFEX | Le Moigne et al. (2009) | parameterizations |
| Soil texture (sand and clay) | Open-ECOCLIMAP | Masson et al. (2003); Faroux et al. (2013) | used to derive soil stratigraphies |
| Soil organic carbon content | Northern Circumpolar Soil Carbon Database version 2 | Hugelius et al. (2013) | smallest geospatial coverage |
| Soil thickness | Gridded Global Data Set of Soil, Regolith and Sediment Thickness | Pelletier et al. (2016) | used to constrain depth to bedrock |
| Snow properties | Global map of snow-climate zones | Sturm et al. (2010) | used to determine minimum and maximum snow densities, and snow aging parameters |

stabilizing effect on the thermal conditions in the soil, temperatures around the freezing point are relatively easy to reproduce
with the model. Overall, comparison between simulated and observed MAGT gives a root mean square error (RMSE) of
$2.21\,\mathrm{K}$, and a slight warm bias of $0.58\,\mathrm{K}$ (Fig. 2b). The agreement with the borehole temperatures is comparable to or better
than reported in previous modeling studies with a similar simulation domain (e.g. Ekici et al., 2014; Willeit and Ganopolski,
2015; Obu et al., 2019). Despite general agreement between observed and modeled thermal states, the analysis reveals some
clear regional differences (Fig. 2a). In lowland permafrost regions (elevation <= $500\,\mathrm{m}$, n=59), the simulated temperatures
differ less from borehole measurements (RMSE=$1.74\,\mathrm{K}$, and bias=$0.17\,\mathrm{K}$) than in mountainous terrain (elevation > $500\,\mathrm{m}$,
n=23) where the deviations are larger (RMSE=$3.10\,\mathrm{K}$) and show a clear warm bias (bias=$1.65\,\mathrm{K}$). This temperature bias is
likely due to the coarse spatial resolution (1°) of the climate forcing data used. Variations in orography are averaged through
the coarse grid resolution, so that topographic climate effects are not accounted for. Since boreholes in mountainous regions are
often located in higher terrain where permafrost occurs (e.g., in Norway), a warm bias is to be expected in a direct comparison.





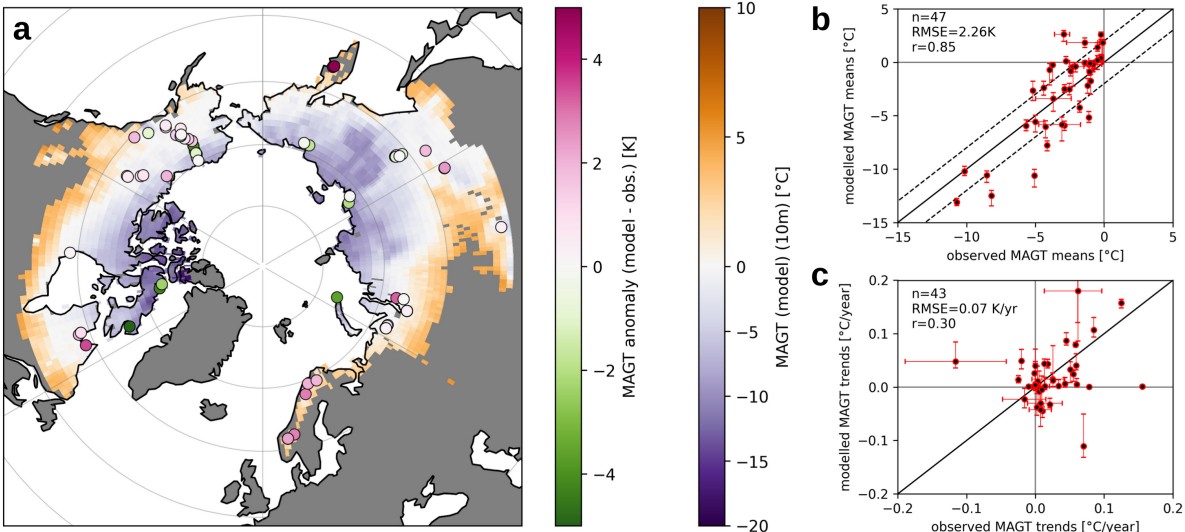

**Figure 2.** The map (a) shows modeled mean annual ground temperatures (MAGT) at 10 m depth from 2007 to 2016. Dots show locations of Global Terrestrial Network for Permafrost (GTN-P) boreholes, with colors indicating temperature deviation between modeled and observed mean ground temperatures (averaged from 2007 to 2016). The scatter plot (b) illustrates the agreement between observed and modeled mean annual ground temperature (MAGT) at 10 m depth with observations lying in the same model grid cell being grouped together. On the y-axis the dots show the mean of the parameter ensemble while the whiskers show the range between the 5th and 95th percentiles. Scatter plot (c) illustrates the agreement between observed and modeled (ensemble-mean) trends in MAGT, each derived from a linear least-squares regression. Observed trends are only included if there are 5 or more years of observations available. Horizontal error bars indicate the range of all observed trends belonging to the same model grid cell. Vertical error bars correspond to the 5th and 95th percentiles of the trends estimated by the parameter ensemble.

At most borehole sites, also the observed temperature trends can be reproduced in sign and magnitude by the simulations (Fig. 2c). Here, the confidence intervals of the observed trends and the range of the simulated trends must be taken into account. Both ranges indicate relatively large uncertainties in temperature trends in particular at those sites showing relatively strong trends (absolute trend values $>0.1\,\mathrm{K\,yr^{-1}}$). On average the simulated trends show a RMSE of $0.07\,\mathrm{K\,yr^{-1}}$. Nevertheless, the comparison between observed and measured temperature trends suggests that the model tends to underestimate observed warming.

### 3.1.2 Active layer thickness

The capability of the model to reproduce the Active Layer Thickness (ALT) defining the depth of the permafrost table is evaluated using field observations of Circumpolar Active Layer Monitoring (CALM) program (Shiklomanov et al., 2012). The dataset comprises a total of $n = 259$ sites within the simulation domain which are located in $n = 94$ different model grid cells.





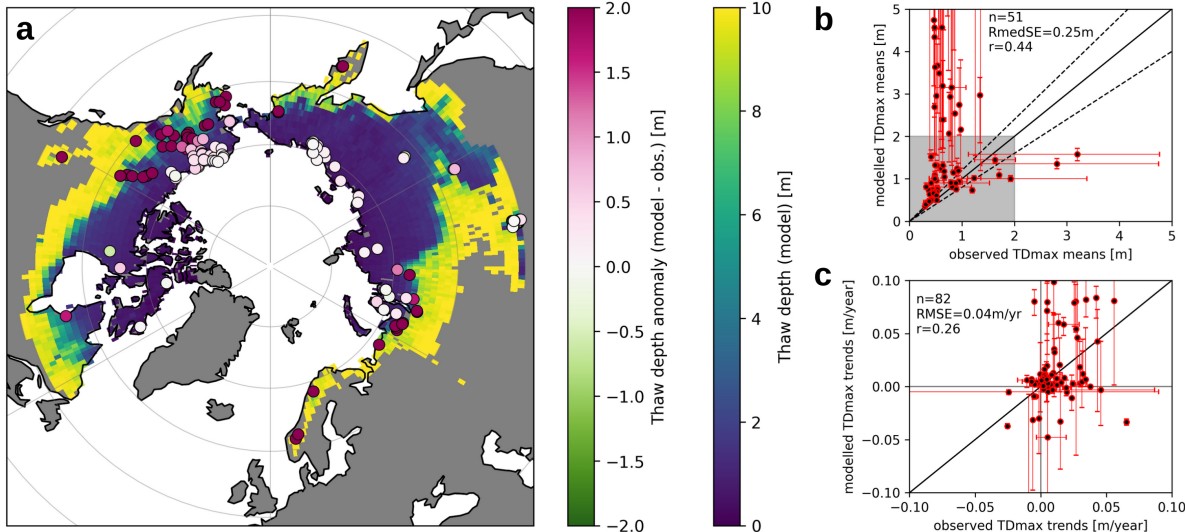

**Figure 3.** Map (a) shows the simulated maximum annual thaw depth (Active Layer Thickness, ALT). Dots show the locations of ALT measurements in the Circumpolar Active Layer Monitoring (CALM) program, with colors indicating the mean deviation between model and observations. Scatter plot (b) compares CALM measurements and modeled ALTs. On the y-axis the dots indicate the mean of the model ensemble with error bars indicating the range between the 5th and 95th percentiles. On the x-axis the dots indicate the mean observed ALT thickness averaging all sites located within the corresponding model grid cell and the error bars show the range of observations if there is more than one observation in the corresponding grid cell. The evaluation metrics given in the upper left corner are for the $n = 51$ points for which both measured and observed ALTs were $<2\,\mathrm{m}$ (gray square). Scatter plot (c) shows modeled and measured trends in ALT for all sites with observations available for 5 or more years. Vertical error bars correspond to the 5th and 95th percentiles of the simulated ALT trends, and horizontal error bars indicate the range of observed trends for grid cells with more than one corresponding measurement sites.

Relatively small deviations ($<0.1\,\mathrm{m}$) between observed and modeled ALTs are found in the northern permafrost regions with thin ALTs ($<1\,\mathrm{m}$) (Fig. 3a). In contrast, large deviations between modeled and observed ALTs (exceeding $2\,\mathrm{m}$) are found for southern permafrost regions except for a few locations in the South Siberian Mountains. On average, the model is found to overestimate the ALTs in comparison to the CALM observations resulting in a high root median squared error (RmedSE=$0.25\,\mathrm{m}$). However, considering the full range of the modeled parameter ensemble (Fig. 3b) reveals a high sensitivity of modeled ALTs

to the simulated parameter range. In particular locations with modeled average ALTs beyond $2\,\mathrm{m}$ reveal very broad ensemble ranges spanning several meters. At a few CALM sites, the model underestimates ALT; however, measurements within the same model grid cell at these sites show very high variability. Given the wide range of simulated ALTs among the members of the parameter ensemble and the fact that CALM sites are specific point observations rather than representative regional averages, it is found that the model at least partially reproduces observed ALTs realistically. It is also important to point out

that ALT measurements with poke probes are susceptible to bias at high ALTs, as gravel or other compact soil material could be falsely misinterpreted as permafrost. We further point out that in southern areas, ALTs are preferentially measured at known



permafrost sites such as peatlands, which are probably not representative of the large-scale picture. In addition, the applied model does not account for thaw subsidence and soil compaction processes, which can significantly reduce ALT, especially in ice-rich regions (e.g. Günther et al., 2015). Therefore, perfect agreement between modeled and observed ALT on a Pan-Arctic

scale is not expected. Nevertheless, the simulation demonstrates the high sensitivity of the simulated ALT to local ground water and ground ice contents, which are subject to large uncertainties and strong spatial variability. While the parameter ensemble simulations can provide insights into the associated model uncertainties, the actual spatial variability of ground water and ice content remains an unresolved challenge.

The comparison between model and observed ALT trends shows a large scatter (RSME=$0.04\,\mathrm{m\,yr^{-1}}$ with maximum values

of $0.10\,\mathrm{m\,yr^{-1}}$) (Fig. 3c). However, the magnitude and sign of ALT trends are captured by the model, especially when uncertainties in the parameters are considered. The wide ranges of the ensemble simulations reveal a high sensitivity of the ALT trends to the ground water and ground ice contents. On average, the model has a tendency to overestimate ALT changes for both ALT growth and ALT shrinkage using the current parameter ranges.

### 3.2 Spatial and Temporal Evolution of the Thermal State of Permafrost

We evaluate the changes in the Thermal State of Permafrost (TSP) over three centuries from 1750 until 2000. We use the pre-industrial period (1850-1900) as the reference, based on which temperature anomalies are calculated for a previous period (1750-1800) representing the later 18th century (L18C) and a subsequent period (1950-2000) representing the more recent historical climate (HIST). The calculated TSP values refer to temporal averages of the ensemble mean.

The TSP anomalies show that the entire Arctic permafrost region is about $0.51\,\mathrm{K}$ colder during the later 18th century

compared to the pre-industrial period (Fig. 4a). The permafrost on the North American continent (defined as 190° E - 310° E) is on average $0.6\,\mathrm{K}$ colder during the L18C with the strongest TSP anomalies found in north-central Alaska which is found more than $1\,\mathrm{K}$ colder during the L18C than during the pre-industrial period. Temperature reconstructions based on tree-ring analyses show a distinct warm period for central Alaska between 1850 and 1900 (Barber et al., 2004), while lake sediment analyses suggest a distinct cold and dry period within the L18C period for the Alaskan Brooks Range (Bird et al., 2009). Cold

air temperatures and low snow depths due to low precipitation explain the cold permafrost temperatures between 1750 and 1800 in north-central Alaska in our simulations. On the Eurasian continent, the differences in the TSP between the L18C and pre-industrial period are found to be slightly smaller. The permafrost in eastern Siberia (defined as 120° E - 190° E) is found to be $0.55\,\mathrm{K}$ colder and in western Siberia (60° E - 120° E) the simulated difference between the L18C and the pre-industrial period is on average only $-0.41\,\mathrm{K}$.

Between the pre-industrial and the historical climate periods, the simulations reveal a general warming of the Arctic permafrost ($+0.7\,\mathrm{K}$, Fig. 4c). Three relative "hotspot" regions with TSP anomalies above $1\,\mathrm{K}$ can be identified in northeastern Canada (Québec: 280° E - 310° E), north Alaska (North Slope), and Western Siberia. The region in Québec shows the strongest TSP anomaly indicating an average warming of $1.39\,\mathrm{K}$ between the pre-industrial and historical periods. At some grid cells the warming exceeds $3\,\mathrm{K}$. This pronounced regional permafrost warming in our simulations is attributed to a warming of air

temperature between 1970 and 1990 and to a shift in the seasonal snowfall distribution with earlier and heavier snowfall in





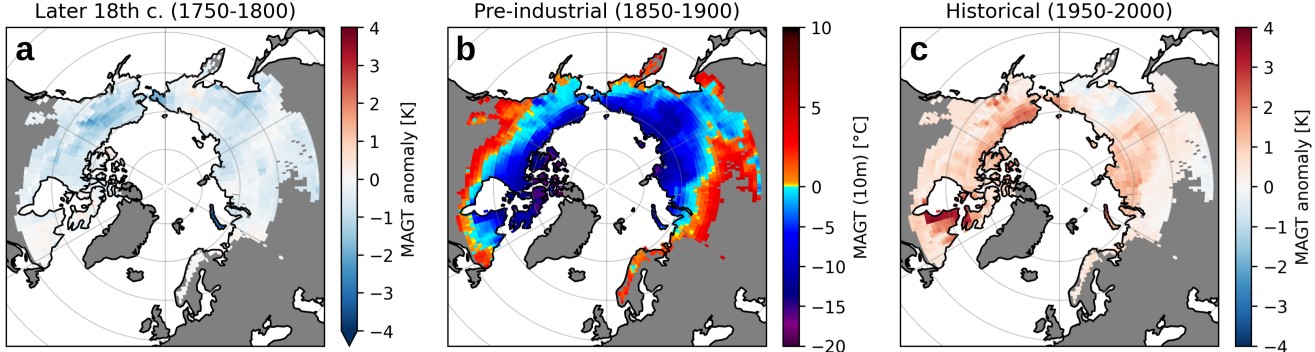

**Figure 4.** Map (a) shows the simulated anomaly of the mean Thermal State of Permafrost (TSP) during the later 18th century (1750-1800) relative to map (b), the mean TSP during the pre-industrial period (1850-1900). Map (c) shows the TSP anomaly of the historical climate period (1950-2000) relative to the TSP during the pre-industrial period.

autumn (October - November). On the North Slope the TSP has also warmed by more than 2 K, while the "hotspot" of warming in western Siberia is much less pronounced, but still more than 1 K over a larger region.

Our simulations reveal pronounced regional differences in the development of TSP during the last 300 years. Atmospheric circulation patterns controlling the seasonal distribution of air temperature and precipitation may be responsible for the ob-
served regional differences. The seasonal interaction of temperature and snowfall and its strong impact on the TSP is well known (Smith et al., 2022). Our long-term simulations over the last three centuries reveal an increased spatial heterogeneity in regional TSP change. During the last decades (1950-2000), permafrost warming is mainly occurring in three "hotspot" regions. However, depending on long-term changes in Arctic atmospheric circulation, regional shifts or the emergence of other "hotspot" regions may be expected.

## 3.3  Spatial and Temporal Evolution of the Active Layer Thickness

Similar to changes in the thermal state of permafrost, we evaluate relative changes in mean Active Layer Thickness (ALT) based on the same centennial periods used to evaluate TSP anomalies. The simulations reveal low to moderate changes in ALT between the L18C and the pre-industrial period (Fig. 5a). ALT changes on the order of 50% are nevertheless simulated for a narrow band on the North American continent (Fig. 5b). The simulated ALTs during pre-industrial times (Fig. 5b) show that the
transition from ALTs less than 2 m to ALTs greater than 10 m is very sharp. This narrow transition band marks the zone of active permafrost degradation and, thus, indicates southern boundary of stable near-surface permafrost. For the Eurasian continent, minor changes (<15%) in mean ALT are simulated between the L18C and the pre-industrial period. Substantial ALT changes are, however, simulated between the historical and pre-industrial periods (Fig. 5c). The simulation show a substantial increase in ALT more than 100% in the narrow zone boundary zone of near-surface permafrost on the North American continent. In
particular northeastern Canada (Québec) and southwest Alaska are affected. Lower but still substantial increase in ALT (25





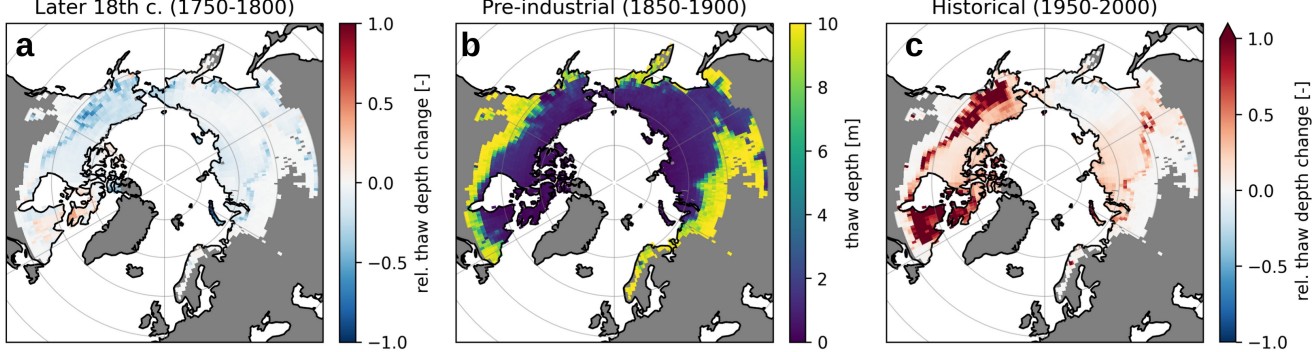

**Figure 5.** Map (a) shows the relative change in mean active layer thickness (ALT) between the later 18th century (1750-1800) and the pre-industrial period (1850-1900). Map (b) shows the mean ALT during pre-industrial period. Map (c) shows the relative change of mean ALT between the historical climate period (1950-2000) and the pre-industrial period.

to 80%) is simulated at the boundary of the near-surface permafrost on the Eurasian continent. The simulations, thus, show a much greater loss of near-surface permafrost on the North American continent than on the Eurasian continent over the past 250 years.

## 3.4 Near-surface permafrost extent

The performed parameter ensemble provides a more detailed insight into the evolution of the near-surface permafrost extent based on the probability of its occurrence. Similar to Obu et al. (2019) we distinguish four zones of permafrost occurrence probability (continuous, discontinuous, sporadic, and isolated permafrost). We consider a model grid cell to contain permafrost if the simulated ALT is less than 3 m (Lawrence et al., 2008). The zoning of permafrost was then done depending on how many members of the parameter ensemble fulfill this criterion (continuous $> 90\%$, discontinuous $50 - 90\%$, sporadic $10 - 50\%$,
isolated $< 10\%$ and $> 0\%$, none $0\%$). The simulations show a moderate increase of total permafrost region extent during the 18th century ($+0.38 \times 10^6 \, \mathrm{km}^2$, Fig. 6a). This increase is despite a slight decrease in the extent of continuous permafrost from 1700 to 1800 ($-0.10 \times 10^6 \, \mathrm{km}^2$) which is outweighed by an increase in discontinuous, sporadic and isolated permafrost extent. The greatest extent of permafrost during the simulation period occurs in 1820 (Fig. 6b), with Alaska and northeastern Canada largely covered by continuous permafrost. Later in the 19th century, the total extent of permafrost begins to decrease slightly
($-0.16 \times 10^6 \, \mathrm{km}^2$). This is driven by a marked decrease in continuous permafrost extent ($-0.77 \times 10^6 \, \mathrm{km}^2$) which is only partly compensated by increases in the non-continuous permafrost zones (about $+0.2 \times 10^6 \, \mathrm{km}^2$ in each zone). During the first half of the 20th century the total permafrost extent remains constant while the continuous permafrost zone continues to shrink ($-0.5 \times 10^6 \, \mathrm{km}^2$ from 1900 to 1950). In 1950 Alaska and northeast Canada have already lost substantial areas of continuous and discontinuous permafrost while the permafrost zones in Eurasia appear largely unaffected (Fig. 5c). The second half of the
20th century is characterized by a more rapid decrease in total permafrost extent ($-0.86 \times 10^6 \, \mathrm{km}^2$), which is driving a similar





loss of the total permafrost region extent ($-0.82 \times 10^6 \, \mathrm{km}^2$). Current permafrost extent (2020) shows substantial area loss in all permafrost zones throughout the permafrost domain (Fig. 6d).

### 3.5 Response of permafrost to climatic events

We note that the beginning of the 19th century as well as the decades around the year 1900 experienced several large volcanic eruptions with impacts on the global climate (volcanic eruptions with VEI$\geq$ 6 during the analysis period: *Unknown* (1808/1809, cf. Guevara-Murua et al. (2014); Timmreck et al. (2021)), Tambora (1815), Krakatoa (1883), Santa-Maria (1902), Novarupta (1912)). These climate events are included in the applied forcing data ((Phipps et al., 2013)) and their impact on permafrost extent can therefore be assessed. Our simulations suggest that the two tropical eruptions at the beginning of the 19th century, as well as the Novarupta eruption in the northern hemisphere (Alaska) are associated with positive effects on the Arctic permafrost extent (Fig. 6). In particular, the zones of continuous, discontinuous, and sporadic permafrost are simulated to expand by several hundred thousand square kilometers after these events. However, these expansions only last for two to three decades before the zones began to decline again (Fig. 6).

### 3.6 Limitations and uncertainties

To keep CryoGridLite efficient enough to run a parameter ensemble simulation ($n = 50$ per grid cell) for the Arctic permafrost region at a spatial resolution of $1°$ and over a total time period of 1400 years, several simplifications and process exclusions are made. The model limitations and the resulting uncertainties in our simulations are summarized below.

- In this version of the model, ground freezing is approximated by an enthalpy-temperature relation of free water (see Appendix A). The implicit scheme used for CryoGridLite can theoretically be applied to arbitrary freezing characteristics, but convergence is not guaranteed and may require under-relaxation (Swaminathan and Voller, 1992). For the simulations and analyses performed, this means that the phase change requires the absorption of the full latent heat content from positive degree-days to increase the active thaw height. If the phase change occured partially below the freezing point temperature, as it would be the case for clay-rich soils, less latent heat from positive degree-days must be invested, allowing for larger thaw depths. Thus, the simulated ALT may be underestimated for soils with high clay content. Furthermore, any freezing point depressions (e.g., occurring in coastal regions due to salt) are not taken into account.

- The current model does not account for groundwater changes, so that the total water-ice content for each ensemble member is assumed to be constant throughout the simulation period. Therefore, the model does not represent soil water fluxes. Thus, the effects of a climatically changing water balance on the thermal state of permafrost are not captured by the simulations. However, uncertainties in soil water and ice contents are represented by parameter ensemble simulations. The limitations imposed by a static hydrology are accepted to avoid the high computational cost of a full surface energy balance, which would require a model forcing that resolves the diurnal cycle and additional uncertain surface parameters.



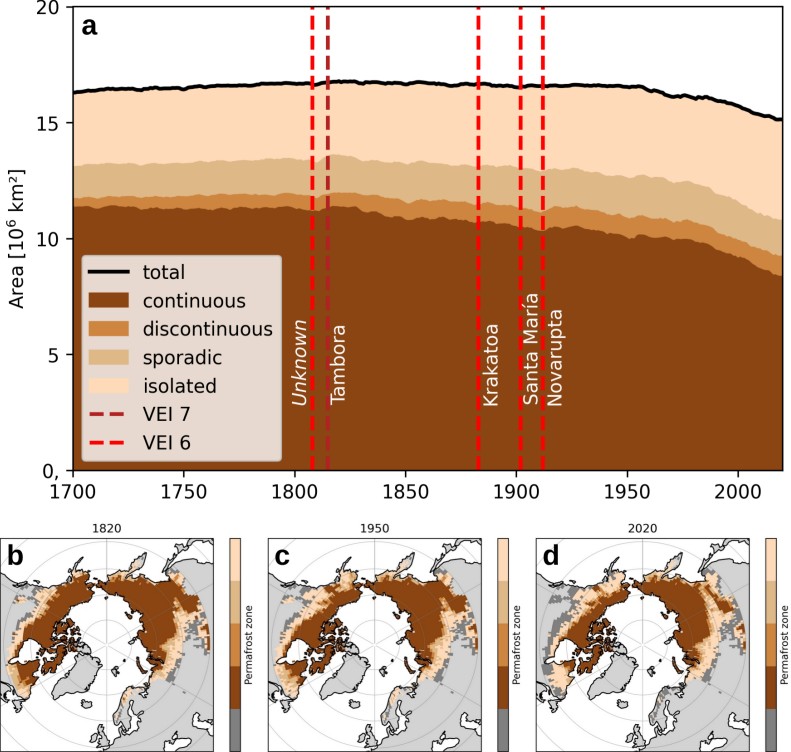

**Figure 6.** Total areas of near-surface permafrost occurrence differentiated according zones of occurrence probability (a) as derived from the parameter ensemble simulations. Dashed red lines mark strong volcanic eruptions events (Volcanic Eruption Index (VEI) ≤ 6) which are represented in applied climate forcing data. The maps permafrost occurrences show specific time slices indicating maximum permafrost area in 1820 (b), permafrost area before accelerated climate warming in 1950 (b), and current permafrost occurrence in 2020 (d).

 – The probabilistic representation of soil and snow properties provides a differentiated insight into the probability of permafrost occurrence. However, the ensemble of parameters used does not necessarily represent the actual local variability of surface and subsurface conditions. Due to lacking data on the probability distributions of the varied parameters, uniform distributions are used. Thus, permafrost occurrence are assessed by plausible parameter ranges, but the parameter ensemble does not necessarily correspond to the spatial permafrost occurrence probability. Furthermore, the simulations performed put a focus on the variability of ground water and ice distributions, and highly variable snow cover properties. In particular, disregarding the variability of sub-grid topography potentially results in underestimating permafrost occurrence in mountain areas (Fiddes et al., 2019).

405 – The current scheme does not account for soil mechanical processes, such as ground subsidence due to pore ice and excess ice melt. Such thermokarst processes can greatly accelerate permafrost thaw regionally (Nitzbon et al., 2020). In addition, the warming effect of surface water such as ponds, lakes, and rivers (e.g. Langer et al., 2016; Juhls et al., 2021;



Ohara et al., 2022) on the thermal state of the underlying and surrounding permafrost is also not considered. Accounting for such non-gradual thaw processes would likely increase the simulated loss of permafrost areas

– There are also processes that can stabilize permafrost, particularly in interaction with the vegetation cover including forests, shrubs, and mosses (e.g. Beringer et al., 2001; Stuenzi et al., 2022; Heijmans et al., 2022). On time scales of centuries to millennia, change in vegetation cover could lead to an expansion and preservation of near-surface permafrost in in the zone of discontinuous and sporadic permafrost where the current model overestimates ALTs systematically. In addition, the increase in permafrost areas due to the retreat of surface ice (glaciers and ice sheets) is not taken into
account.

Despite these limitations, we find that CryoGridLite provides a solid and efficient basis for reproducing permafrost thermal dynamics realistically. Nevertheless, the identified shortcomings give rise to future model improvements. Operational parameterizations for local and regional permafrost models already exist for most of the processes not considered in the current model version (Nitzbon et al., 2019; Stuenzi et al., 2022; Westermann et al., 2022). Simplified and numerically optimized
parameterizations can be derived from these and implemented in CryoGridLite. In particular, the ability of CryoGridLite to compute many instances of a grid cell allows the implementation of multi-tile approaches to represent sub-grid processes and their lateral interactions (Martin et al., 2021; Nitzbon et al., 2021).

## 4   Conclusions

In this study, an efficient numerical permafrost model is presented that bridges the gap between reduced-order permafrost
schemes used in intermediate complexity climate models and very detailed permafrost process models. The CryoGridLite model is tailored to enable parameter ensemble simulations of ground thermal dynamics covering the entire Arctic permafrost region and timescales beyond centuries. Despite limited process inclusion, the model is capable of estimates of the thermal state of permafrost (RMSE=2.21 K) and its current warming rate. Taking into account possible biases caused by neglecting subgrid variations in topography due to coarse spatial resolution of the climate data used, even higher accuracies are obtained
(RMSE=1.74 K). The model is also able to provide a realistic estimate of Active Layer Thickness (ALT), especially in the zone of continuous permafrost. Large uncertainties in ALT simulations are found in the zones of discontinuous to isolated permafrost attributed to uncertain groundwater and ground ice contents.

The simulations performed show spatially heterogeneous warming of the permafrost during the last 250 years. Three "hotspot" regions characterized by particularly strong warming (>1 K) since industrialization (1900) were identified. Changes
in ALT are simulated to occur mainly along the boundary between continuous and discontinuous permafrost. In particular, permafrost on the North American continent has been affected by a substantial increase in ALT (>100%) since industrialization whereas much smaller changes in ALT are simulated for the Eurasian Arctic permafrost. Generally, the North American continent is characterized by intense permafrost thaw while Eurasian permafrost appears to have been less affected over the past 250 years.




The near-surface permafrost extent in the Arctic has changed significantly over the past 250 years. All Arctic permafrost zones combined have lost about 12% of their area since 1850, with the most affected zone of continuous permafrost showing an area loss of about 20%. A greatly accelerated decline in permafrost extent has occurred after the 1950s, with a loss of area of about $-1.36 \times 10^5 \, \mathrm{km^2 \, dec^{-1}}$. It was found that climatic events caused by volcanic eruptions affect permafrost extent only for a very limited duration of a few decades.

Despite limited process representation compared to more complex permafrost process models, we conclude that CryoGridLite provides important insights into the long-term evolution of the thermal ground regime on large spatial scales. In particular, the model's ability to link multiple global datasets using a probabilistic ensemble approach allows CryoGridLite to deal with highly uncertain ground and snow properties. Future simulations could cover even larger time scales to investigate the formation of permafrost as a result of transient climate conditions from the Pleistocene to the present.

*Code and data availability.*  The CryoGridLite model code is available from https://zenodo.org/record/6619537. The input data required for pan-Arctic simulations are available from https://zenodo.org/record/6619212. Model output used for the results presented in this article is available from https://zenodo.org/record/6619260.

**Appendix A: Method details**

Volumetric enthalpy, $H_v$, as a function of temperature, $T$, is defined as:

$$H_v = TC_v(T) + Lf(T) \tag{A1}$$

where $C_v(T)$ $[\mathrm{J\,m^{-3}\,K^{-1}}]$ is the temperature dependent volumetric heat capacity of the medium, $L$ $[\mathrm{J\,m^{-3}}]$ is the volumetric heat of fusion of water, and $f(T)$ $[\mathrm{m^3\,m^{-3}}]$ is the freezing characteristic curve which defines the relationship between temperature and volumetric liquid water content. The free water freezing characteristic defines the liquid fraction of water in terms of volumetric enthalpy:

$$f_{wl}(H_v) = \begin{cases} \theta & H_v > L\theta \\ \frac{H_v}{L} & 0 \le H_v \le L\theta \\ 0 & H_v < 0 \end{cases} \tag{A2}$$

where $\theta$ is the total water content. Temperature can be determined via the corresponding inverse enthalpy function:

$$H_v^{-1}(H_v) = \begin{cases} \frac{(H_v - L\theta)}{C_v} & H_v > L\theta \\ 0 & 0 \le H_v \le L\theta \\ \frac{H_v}{C_v} & H_v < 0 \end{cases} \tag{A3}$$



While the derivative of the enthalpy function $\frac{dH_v}{dT}$ beyond the critical enthalpy range where a phase change occurs ($H_v < 0$ or $H_v > L\theta$) can simply be equated to $C_v$ (see Eq. A1), within this range it would technically be infinite (see A2). In this case, a numerically

convenient workaround is to simply set it to a very large value, e.g. $\frac{dH_v}{dT} \approx 1 \times 10^9 \, \mathrm{J\,m^{-3}\,K^{-1}}$.

**Table A1.** Overview of the heat capacity and thermal conductivity values used for the individual soil constituents. The values are based on Hillel (1998)

| Soil constituent | Volumetric heat capacity [$\mathrm{J\,m^{-3}\,K^{-1}}$] | Thermal conductivity [$\mathrm{W\,m^{-1}\,K^{-1}}$] |
|---|---|---|
| Water | $4.2 \times 10^6$ | 0.57 |
| Ice | $1.9 \times 10^6$ | 2.2 |
| Organic | $2.5 \times 10^6$ | 0.25 |
| Mineral | $2.0 \times 10^6$ | 3.8 |
| Air | $1.25 \times 10^3$ | 0.025 |

*Author contributions.* M.L. and S.W. conceptualized the study, M.L. developed the core of CryoGridLite model, and performed the major analysis, J.N. performed and analysed the ensemble parameter simulations and provided the figures. B.G. performed the evaluation of the model against numerical solvers and supported the technical model implementation. L.-M.A. implemented the web-map for out visualization. All authors contributed to writing the manuscript and supported the evaluation of the results.

*Competing interests.* There are no competing interests.

*Acknowledgements.* This work was supported through a grant by the Federal Ministry of Education and Research (BMBF) of Germany (No. 01LN1709A, Research Group PermaRisk) awarded to Moritz Langer. Brian Groenke acknowledges the support of the Helmholtz Einstein International Berlin Research School in Data Science (HEIBRiDS). Sebastian Westermann acknowledges the support of Permafrost4Life (Research Council of Norway, grant no. 301639) and ESA Permafrost_CCI (https://climate.esa.int/en/projects/permafrost/).



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
