# Peer review of "The evolution of Arctic permafrost over the last three centuries"

_EGUsphere, 2022_

## Editor Comment (EC1)

Dear Moritz Langer and colleagues,

Thank you for your submission to The Cryosphere (TC). We noted that you asked to submit an updated version of your manuscript before starting the review process – but unfortunately this is not possibly at this stage, given that you uploaded your manuscript as a preprint on EGUsphere (we double checked with the editor-in-chief and the Copernicus team). If you think the changes you performed are important to mention at this stage, I invite you to provide information on the changes as a comment in the discussion. Otherwise, you are invited to just include these when sending your revised manuscript addressing the reviewer comments. I am sorry for this inconvenience caused by the new setup with preprints being available on egusphere. Similarly, I was not able to provide comments before the start of the discussion phase ('the quick access review'). I nevertheless provide them here, and invite you to incorporate these when addressing the reviewer comments.
* * *
1. Originality (Novelty):

In this manuscript, the authors present a new model, CryoGridLite, which they extensively evaluate and utilize to simulate the evolution of Arctic permafrost over the past three centuries. The long-term focus of the study is interesting and is made possible through the fact that CryoGridLite relies on some simplifications and parameterizations that make it faster to run than more complex permafrost models. This allows the authors to make some claims on the long-term evolution of permafrost, thereby highlighting some clear regional differences that may in part not yet be (well) known.

2. Scientific Quality (Rigour):

CryoGridLite seems to find the weak spot between complex permafrost models and (very) strongly simplified parameterizations. The approximations described in the manuscript seem relevant, and will be further judged by reviewers. The elaborate model evaluation (section 3.1) is really helpful and provides confidence in the shown results, while at the same time acknowledging some of the limitations (highlighted in section 3.6).

3. Significance (Impact):

The impact of this study may be two-fold. First, the new insights on the long-term evolution seem to be quite important for the community. Second, the newly presented model (including the data needed to represent the results as presented here – excellent) is likely to be used for many other applications in the future.

4. Presentation Quality:

The manuscript is very well written and easy to follow – also for someone without any background in permafrost science. The figures are also very clear, visually appealing, and support the story well.
* * *
List of (mostly very minor) comments that I noted when going through your manuscript:

- At first it was not clear to me that your new model is CryoGridLite. To highlight this, and for this manuscript to also serve as the reference for this model in the future, I suggest:
    - mentioning the name of the model in your abstract. You could even consider adding it in the title, although this entirely up to you.
    - Explicitly mention that it is your new model in the methods section. Now, the model is never named, and you immediately start with 'CryoGridLite largely…'. Maybe something along the lines of 'Our newly developed CryoGridLite model largely…'?
- l.66: add a space between the two sentences + 'permafrost' instead of 'permarost'
- l.71: To avoid having two subsequent sentences starting with 'In addition', suggest removing this from previous sentence (in l.69)
- l.109-110: when mentioning the stability of the models. Does the fully implicit model approach require such a small time step as well? And if so, is the requirement for the others even more constraining (e.g. (sub-)hourly)? Just a question out of interest – that could potentially be answered by providing slightly more info here
- l.141: "simultaneously with the ground": at the same time as the calculations performed for the ground? Maybe reword to make this clearer.
- l.158-159: the lowest grid cells: at which depth are these occurring?
- l.160: when reading this sentence, I wondered if this means that you can maximally model a snow-depth of 2 m (which was later confirmed when seeing the values in Table 1). What is the reason for this – and is this a limitation? Most likely not, but maybe good to mention why.
- Caption of figure 1: "This way a wide…" → "Through this approach, a wide"
- Table 1 caption: 'which were varied' → "which are varied"? to be consistent with the text? Same comment applies to the second sentence.
- l.221: "which are drawn"?
- l.239: why do you rely on ERA-Interim and not ERA-5? Not suggesting this needs to be changed, but would be good to clarify, since the latter is by now in most cases the reference (or is this not the case in the permafrost community)?
- Figure 2 and 3 caption: replace '5' by 'five' (generally, it is recommended to write out the numbers smaller than ten)
- l. 305: using the term 'evaluate' is definitely correct here, but may be a bit confusing, since you have just done what you refer to as the evaluation in the previous section (3.1). Maybe you want to make this distinction clearer and refer to 'analyze' here?
- l.311-312: "which is found more than" → "which is found to be more than…"
- l.341: "…indicates southern boundary…" → "…indicates the southern boundary…"
- l.343-344: "the simulation shows a substantial increase in ALT of more than…"
- Figure 5 caption: here you refer to the active layer thickness (ALT), while in the figure itself this is referred to as the that depth. Any reason for using two different terminologies? If not, would suggest having this consistent here (also for other occurrences in text)

- l.353: did I understand it correctly that if the ALT is more than 3 m, this is not considered to be permafrost then?
- l.370: introduce VEI here? Only done in caption of Figure 6 now
- l.372: remove the double brackets for the reference
- l.377: "begin" to be consistent with the rest of the sentence?
- Caption figure 6:
  - VEI >= 6, right?
  - Suggest rewording the last sentence to: "The maps show permafrost occurrence at specific time…"
- l.472: suggest rewording to: "…capable of estimating the thermal…"
- Author contribution:
  - What is "the web-map"? Something online or shown here?
  - And similarly, not sure I understand what "out visualization" is?
- l.522: replace the '∼' in title of paper
- l.573: 'B ayesian' → 'Bayesian' (two occurrences)

Best regards,
Harry

---

## Author Comment (AC2)

**Reply letter to reviewer 1**

*We thank the reviewer for the interest in our study and appreciate the comments and critical remarks, which have helped to improve our manuscript. We hope to answer all issues raised satisfactorily in the following response. Please note that our replies are in italics and changes made in the manuscript are highlighted in* **bold***. Updated figures are shown, but may not be exactly as shown later in the revised version due to ongoing graphical editing.*

General Comments

The manuscript by Langer et al. utilizes a numerical model to simulate the evolution of Arctic permafrost for the last three centuries (1750-2000). The response to changes in air temperature of the thermal state of permafrost and active layer thickness are simulated and conclusions have been made regarding the impacts on permafrost distribution. These types of simulations are of interest and can help to provide better understanding future evolution of permafrost conditions. The paper is interesting and is within the scope of "The Cryosphere" and would be of interest to permafrost scientists. I have no major issue with the numerical thermal model that is utilized. However, there are some issues that should be addressed regarding model inputs and validation for the MS to be acceptable for publication.

Values for thermal properties have been provided in Table A.1. Only one value is provided for the mineral component of soil, and it is unclear how this value was chosen – is this considered an average value? There is a large difference between the thermal conductivity of clay minerals and quartz for example (2.92 vs 8.80 $Wm^{-1}K^{-1}$ – values from Williams and Smith 1989) which will be an important factor in the thermal response.

*We thank the reviewer for this comment and agree that the thermal properties of the soil constituents, in particular the thermal conductivity of minerals, introduce additional uncertainty into the evolution of the ground thermal regime. For simplicity and in the absence of detailed information on the occurrence of soil and bedrock minerals, we are in fact assuming an average value of 3.8 [W K-1m-1] for the thermal conductivity of the mineral fraction of the soil. This value is within the range of values (2.9 - 8.8) $Wm^{-1}K^{-1}$ given by Hillel (1982). We argue that in the near surface ground layer the uncertainties associated with the soil, sediment, or rock porosity and their water saturation induce considerably larger uncertainties in the ground thermal regime because there the conductivity varies not only by a factor of three but by two orders of magnitude. With the parameter ensemble simulations we aim to account for this variability of the near surface layer layer, but we neglect uncertainties and variabilities of deeper ground layers. To clarify this, we add the following explanation in the manuscript:*

**We emphasize that the parameter ensemble approach is limited to representing the variations in the parameters controlling the thermal dynamics of the ground near the surface. Variations in the thermal properties of the deep ground (bedrock) resulting from variations in geological conditions and mineralogical composition are neglected. For the bedrock layer, we assume a thermal conductivity composed of an average mineral thermal conductivity (\qty{3.8}{\watt\per\meter\per\kelvin}) and the thermal conductivity of the water/ice-saturated pore space, where the pore space varies between soil porosity and 1\%.**

It also is not clear what value is used for bedrock and whether it varies with the mineralogy of the rock (the quartz content will be an important factor). There does not appear to be any information provided regarding the source of information on the type of bedrock or inclusion of bedrock stratigraphy in the model.

*As for the upper soil layers, we did not vary the thermal conductivity of the mineral fraction according to the mineralogy. For the bedrock, we assumed a thermal conductivity composed of the average mineral thermal conductivity (3.8 Wm^-1K^-1)) and the thermal conductivity of the water/ice-saturated pore space, with the pore space varied within the parameter ensembles between the soil porosity and 1%. In this way, a large range of uncertainty is accounted for in the ensemble simulations performed. However, we note that we may underestimate the thermal conductivity in the case of very high fractions of quartz in highly consolidated rocks. To clarify this limitation in our parameterization, we have added the following statement:*

**We note that the thermal properties of the subsurface can have a substantial effect on the ground temperature profile. The analysis therefore focuses on the thermal state of the upper ground ($\le$ \qty{10}{m}), where uncertainties in the composition of the upper ground layer are assumed to have a disproportionately larger influence. When studying the thermal state in deeper ground layers, the differences in mineralogy should be taken into account.**

Validation of the ability of the model to reproduce the ground thermal regime and temperature trends is mainly done through comparison to borehole temperatures at 10 m depth for 2007-2016 extracted from Biskaborn et al. (2019). Deviations of up to 2K are reported which does seem rather large. It is not clear why the 10 m depth was chosen, and it should be noted that the values given in Biskaborn et al. (2019) are only provided for one depth, i.e. zero annual amplitude (ZAA) or the measurement depth closest to it. In some cases ZAA was much deeper than the measurement depth including at some sites where the measurement depth is 10 m. Although temperatures at 10 m depth may show little seasonal variation for some sites (eg. forested warm permafrost sites), for other sites there may be considerable seasonal variation and it may be more difficult to evaluate long-term trends. Only the trend over a 10-year period has been utilized for validation and it is not clear why other information on trends over longer periods of time has not been utilized or why other evidence of permafrost evolution during earlier time periods has not been considered in the model evaluation.

*We first have to apologize for the unclear description of the validation procedure regarding observation and model depth. We have to clarify that we indeed used the same depth from the modeled profile as given by observations. In Fig. 2a, the background map shows the modeled mean annual ground temperature (MAGT) in 10 m depth, while the temperature deviations indicated as coloured circles are given for the depth of the model grid which is closest to the depth of the observations. The same is true for the scatter plots in Fig. 2 b,c. This is now noted in the caption of Fig. 2:*

**The map (a) shows modeled mean annual ground temperatures (MAGT) at 10 m depth averaged over the decade from 2007 to 2016. Circles indicate temperature deviations between the observed ground temperatures of the Global Terrestrial Network for Permafrost (GTN-P) boreholes and the modeled temperatures at the depth of the subsurface grid closest to the depth of the respective observations. The scatter plot (b) illustrates the agreement between observed and modeled mean annual ground temperature (MAGT) with observations lying in the same model grid cell being grouped together.**

*Regarding the observed deviations of up to 2 K, we would like to point out that these are relatively small. As stated in the manuscript, the observed deviations are even slightly smaller than in other (state-of-the-art) pan-Arctic modeling studies (e.g., Obu et al., 2019). We would like to stress that in this evaluation, borehole data (which are point measurements in a very heterogeneous landscape) must be compared with coarse-resolution model output data (1x1° grid cells). We partly addressed subgrid-scale heterogeneities with the parameter-ensemble simulations, which indeed resulted in typical spreads of the modeled MAGT of 1-2 K. While our parameter-ensemble approach is thus able to address MAGT variations due to the stratigraphy and snow height, it is limited in its capability to factor in meso-scale topographic effects which can lead to substantially different temperatures at the surface, e.g. in mountainous and hilly terrain. We have extended the discussion of this limitation of our approach as follows:*

**Furthermore, our simulations focus on the variability of ground water and ice distributions, as well as highly variable snow cover properties. While our parameter-ensemble approach was successful in reproducing MAGT variations due to variability in stratigraphy and snow height, it has limitations in accounting for meso-scale topographic effects. These effects are found to result in substantial temperature deviations between observation and simulation in mountainous and hilly terrain.**

*We agree that extending the validation of our simulations over a longer time period is important to demonstrate the model's ability to reproduce long-term changes in the thermal state of permafrost. In the revised version of the manuscript, we have incorporated a more comprehensive validation approach, specifically focusing on borehole measurement sites where long-term data (over 10 years) are available. Please also see the responses to the following comments.*

Trends in permafrost temperature over longer periods, up to 4 decades, are reported for several sites in various publications including the annual State of Climate reports published in BAMS (most recent Smith et al. 2022). Consideration of longer time periods for model validation is important given that rate of permafrost temperature change has varied over time from the latter few decades of the 20[th] century to the present as shown in for example Romanovsky et al. (2010, 2017); Smith et al. (2010, 2022).

*We agree with the reviewer that the validation period covers only a relatively short period of 10 years, whereas our goal is to use the model to simulate long-term changes. We have therefore sought to include more validation data, as suggested by the reviewer. In the revised version of the manuscript, we now incorporate long-term measurements as recently published by e.g. Smith et al. (2022).*

*A more detailed model evaluation is conducted for selected sites where long-term borehole measurements spanning periods of more than 10 years were available. We compare air and ground temperature observations from three sites in Canada (Salluit, Quaqtaq, and Ellesmere Island) \citep{allard2020borehole} and one in Russia (Urengoy) \citep{smith2022} to the corresponding model results from the nearest grid cell within the model domain. For the sites in the Quebec region (Salluit and Quaqtaq), the measured air temperatures systematically show a difference of about \qtyrange{1}{2}{K} compared to the ERA interim forcing. This can be well explained by small scale variability e.g. due to orographic effects or proximity of the measurement sites to the coast. For both sites, the observed mean annual ground temperatures (MAGT) at \qty{10}{m} depth during the 2000s and 2010s are very well within the modeled ensemble range and even agree well with the model mean, especially for Quaqtaq, suggesting that the model ensemble captures site-level conditions despite a bias in the forcing data. For a depth at \qty{20}{m}, additional borehole observations are available for the late 1980s and early 1990s \citep{allard1995recent}. The time series of measurements, however, is not long enough to confirm the pronounced negative temperature anomaly simulated before 1980 for Quaqtaq. For a high Arctic site on Ellesmere Island (Fig. Appendix c), the long-term observations and simulations also appear to agree reasonably well, although the natural variability is not fully captured by the spread of the model ensemble. The long-term measurements from Urengoy (Fig. Appendix d) in Russia show very cold borehole temperatures, and only the coldest simulations in the ensemble show such low temperatures. Another borehole is located in the immediate vicinity which shows much warmer temperatures but has a shorter time series.*

[Figure]

***Figure 1 Appendix:** Model evaluation based on long-term borehole temperature and near-surface air temperature measurements in different regions. (a) Quaqtaq, (b) Salluit in the Quebec region, (c) a high Arctic region on Ellesmere Island, and (d) Urengoy in northern West Siberia, Russia. If multiple borehole measurements are available, they are indicated separately.*

There have also been studies that have compared recent observations of permafrost occurrence and thaw depths to measurements made 4-6 decades earlier (e.g. James et al. 2013; Holloway and Lewkowicz 2020). These observations provide additional information on permafrost evolution, particularly in the southern portion of the permafrost region, that could be compared to model results.

*We thank the reviewer for these suggestions and have included these studies in the discussion of the model's performance in representing long-term changes in thaw depth (see following response).*

There are also studies that use proxy data to consider the evolution of permafrost over the last 6000 years (see for e.g. Treat and Jones 2018). These studies show that permafrost particularly in the current discontinuous zone formed fairly recently, during the Little Ice Age. The latter portion of this cold period overlaps with the 1750-1800 period considered in the results presented in the MS and these proxy data could also be used in the evaluation of model performance. It should also be noted that permafrost that formed during the Little Ice Age persists in peatlands (e.g. James et al. 2013; Holloway and Lewkowicz 2020).

*We also thank the reviewer for these suggestions, which we have considered in further model evaluation. There is very little overlap between the time periods examined in this study and those examined by Treat and Jones (2018). Nevertheless, we gratefully take up the point about permafrost distribution at lower latitudes to further discuss the limitations of our model, which clearly lacks the representation of dynamic vegetation. The evolution of permafrost in dynamically growing peatlands and its resilience to warming is not adequately represented in our simulations. In the revised version this limitation is outlined as:*

***However, we clarify that ensemble simulations cannot capture the complex effects of a dynamically evolving and responding vegetation cover. In particular, in warm permafrost regions where permafrost occurrence is very sensitive to organic surface layer properties \citep{james2013decadal, holloway2020half}. This could be problematic in accurately representing permafrost evolution in peatlands, particularly in the lowest latitude regions where permafrost is reported to be an ecosystem-protected legacy of the Little Ice Age \citep{treat2018near}.***

The authors present estimates regarding loss of permafrost, including loss of continuous permafrost. However, only the extent of near-surface permafrost (upper 3 m) is considered – essentially only considering a change in thaw depth. In the continuous permafrost zone where permafrost is several 10s to 100s of metres thick, loss of permafrost in the upper 3 m does not really provide a characterization of the lateral extent of permafrost. Justification of the loss of continuous permafrost would therefore be rather difficult.

*We thank the reviewer for this comment and we agree that the terminology we used to describe the permafrost zones and our analysis were not consistent. Therefore, we have decided to avoid using terms such as continuous, discontinuous, etc. Instead, we now refer to the probability of permafrost occurrence based on the fraction of ensemble members fulfilling a criterion for permafrost presence/absence. In addition, we now diagnose the presence/absence of permafrost based on two different criteria. The first one, which was included already in our initial submission, focuses on near-surface permafrost, and defines near-surface permafrost to be absent if the maximum annual thawed ground depth in the upper 10m of the subsurface exceeds 3m. The second diagnostic defines permafrost to be present if some ground is frozen year-round within the upper 10m of the ground (see Figures below). It turns out that while the diagnostic for near-surface permafrost is more sensitive to climatic warming in recent decades, the diagnostic for deeper permafrost shows a response to short term cooling as associated with volcanic eruptions. Combined, these diagnostics provide a more complete picture of the simulated dynamical evolution of ground thermal conditions than criteria based on ground temperatures in a certain depth.*

*In the revised manuscript, we have added a new subsection in the Methods, defining the different diagnostics we used to evaluate and analyze our simulation results. In addition, we revised Figure 6 which now allows for an improved picture of the overall temporal evolution of (near-surface) permafrost over the study period.*

[Figure]

[Figure]

**Figure 6. Total areas of probability of near-surface permafrost occurrence according to two different diagnostics focusing on (a) 3 m and (b) 10 m ground depth. The zones of occurrence probability are derived from the parameter ensemble simulations. Horizontal lines mark areas of permafrost regions as delineated by Brown et al. (1997) and Obu et al. (2019) for the early 21st century. Dashed red vertical lines mark strong volcanic eruptions events (Volcanic Eruption Index (VEI) ≤ 6) which are represented in applied climate forcing data.**

Additional Comments

L5 – Trends in what? Active layer thickness?

*We agree that the formulation was unclear and incorporated that sentence into the previous one.*

L9-12 – See comments above regarding interpretation of results with respect to lateral extent of permafrost.

*As explained in our response to the major comment above, we revised the terminology regarding the designation of different types and zones of permafrost to be more consistent with the actual model output. In addition to a model output diagnostic on near-surface permafrost (i.e. permafrost within the upper 3m, as this is conventionally considered in large-scale modeling (Koven et al., 2012, Burke et al., 2020) and for the quantification of carbon pools (Hugelius et al. 2014)) we have added a diagnostic evaluating the permafrost occurrence within the upper 10m.*

**For the model evaluation and analysis of the evolution of permafrost over the last three centuries, we apply different key diagnostics representing the thermal state of permafrost and the active layer. For the ground temperatures, we focus on the mean annual ground temperature in a depth of $10\,$m (MAGT$_{10m}$), but we take additional depths into account for the sites at which we compare our model results with borehole measurements. To assess the state and changes in the active layer, we diagnose the maximum annual thawed ground depth [\qty{}{m}] within the upper \qty{10}{m} of the subsurface (TD$_{10m}$), which corresponds to the active layer thickness (ALT) under stable permafrost conditions.**

**To assess the circumpolar extent of (near-surface) permafrost, we establish two criteria to distinguish the presence or absence of permafrost for a given year and ensemble member. Near-surface permafrost is defined to be absent, if the thawed ground depth is exceeding \qty{3}{m}. Note that this criterion does not exclude the presence of permafrost, particularly in depths exceeding \qty{3}{m}. The corresponding probability of near-surface permafrost occurrence is defined as the fraction of ensemble members where the criterion for absence is not fulfilled:**

**Equation**

**Similarly, we diagnose the occurrence probability of permafrost within the upper $10\,$m of the subsurface as the fraction of ensemble members where some ground within the upper \qty{10}{m} is frozen throughout the entire year:**

**Equation**

**We note that these criteria deviate from the formal definition of permafrost, which is tied to maximum annual ground temperature. However, the applied diagnostics provide a nuanced insight into the dynamic evolution of permafrost conditions over a deeper volume in response to short- and long-term climatic changes. While the first criterion is often used in global permafrost modeling to examine the impacts of climate warming on the permafrost carbon pool, the second criterion allows for the consideration of temporary frozen soil layers down to a depth of 10 meters.**

*Koven, C. D., Riley, W. J., and Stern, A.: Analysis of Permafrost Thermal Dynamics and Response to Climate Change in the CMIP5 Earth System Models, Journal of Climate, 26, 1877–1900, https://doi.org/10.1175/JCLI-D-12-00228.1, 2012/10/01.*

*Burke, E. J., Zhang, Y., and Krinner, G.: Evaluating permafrost physics in the Coupled Model Intercomparison Project 6 (CMIP6) models and their sensitivity to climate change, The Cryosphere, 14, 3155–3174, https://doi.org/10.5194/tc-14-3155-2020, 2020.*

L18 – Models presented in Chadburn and Obu are equilibrium models, so the permafrost distribution determined is not necessarily representative of current conditions.

*In general, we agree that the equilibrium models might not capture the present-day state accurately. However, the product of Obu et al. is the most recent estimate of (near-surface) permafrost extent available. Chadburn et al. furthermore provide estimates of equilibrium permafrost extent at different warming levels. Combined, we think that these references are appropriate to justify our general introductory statement that permafrost is (still) the largest non-seasonal component of the cryosphere.*

L22-23 – There are other references regarding the link between deeper temperatures and past climate. One of the earliest is Lachenbruch and Marshall (1986).

*We fully agree with the reviewer that there are numerous earlier investigations demonstrating this link including the one suggested by the reviewer. In the revised manuscript, we have included the proposed study in addition to other early and contemporary studies (without claiming completeness).*

*(e.g. Lachenbruch and Marshall, 1986; Allen et al., 1988; Osterkamp and Gosink, 1991; Harrison, 1991; Kneier et al., 2018).*

L23-33 – Another paper that considers permafrost that has survived over glacial-interglacial cycles is Froese et al. (2008). It should also be noted that the glacial history is not only important from a climatic perspective with respect to permafrost evolution, it also an important factor in ground ice conditions (see for example O'Neill et al. 2019) including the occurrence of buried ice. It is also related to sea level changes which influence ground ice conditions and evolution of permafrost thermal state in coastal areas and regions below the marine limit.

*We agree that these factors are also important for permafrost history and* extended *the statement in the introduction accordingly.*

L62 – What about uncertainties in bedrock properties?

*The residual water/ice content in the bedrock was varied randomly in our simulations, but we did not vary other bedrock properties systematically, so that we did not mention this explicitly here. Please find a more detailed response above.*

L75 – Is there a bedrock module?

*No, the bedrock is not treated* separately*, but together with the soil atop. For the bedrock, we assumed higher mineral fractions than in the soil which affects its thermal* properties*. To clarify this, we renamed the module for the subsurface thermal dynamics "ground module".*

L101 – Is excess ice considered or only pore ice?

*Here, we only considered the existence of pore ice up to the soil porosity obtained from the input dataset for soil properties. However, we considered the effect of excess ice on the ground heat budget in a follow-up study using the same model (Nitzbon et al., 2022).*

*To clarify this point we have changed to concerning sentence to:*

**For each grid cell the ground thermal properties such as thermal conductivities and heat capacities are calculated based on the actual ground composition defined by the volumetric contents of organic, mineral, pore water, and pore ice. We note that the thermal properties of excess ground ice are not considered here, but are considered in a companion study by Nitzbon et al. (2022) using the same model.**

L122 – Snow cover exhibits much local variability due to for example topography, vegetation, exposure to wind. Is this considered?

*Our model incorporates a simple snow scheme (Strum et al., 2010) that does not explicitly account for modifications to the snowpack, such as wind compaction, lateral redistribution, and the presence of vegetation. Instead, the thermal characteristics of the snow in the model are mainly determined by its initial bulk density further compacting the snowpack age. Initial snow density as well as compaction parameters are set according to empirical regional factors (Strum et al., 2010). Although we acknowledge the limitations of this approach, we did take steps to account for the local variability of the snowpack. Specifically, we set a maximum height for the snowpack based on a randomly chosen value within the parameter-ensemble simulations. This allowed us to represent exposed areas where snow is easily blown away, as these were represented by ensemble members with a lower maximum height. While this does not capture all of the complexities of the snowpack, it does allow us to capture some of the local variability in our model. This is explained in more detail in the manuscript on . 144ff.*

*However to clarify the limitations earlier we have added the following paragraph to the method description:*

**The snow scheme used in this study has limitations in reflecting the local and complex thermal properties of snow. Specifically, the scheme does not explicitly account for processes such as wind compaction and lateral redistribution of snow due to local topographic variability, nor does it explicitly represent local interactions with vegetation. The empirical parameterization used accounts only for regional variability in bulk snow density and its compaction behavior.**

L148 – Does the observational data support setting snow depth to zero in August?

*We thank the reviewer for this comment. Perennial snow and ice fields, which result from the buildup of multi-annual snow, are observed in the Arctic. Satellite observations indicate that the total terrestrial area covered by perennial snow and ice is approximately 5e5 km² (Fontana et al., 2010), which corresponds to approximately 3% of the current permafrost area. These features are predominantly found in mountainous regions, primarily as glaciers or ice fields at specific locations with high snow accumulation. Although we recognize the importance of these areas, they are relatively small compared to the entire permafrost domain. Moreover, their occurrence is strongly influenced by micro- and mesoscale snow redistribution processes, which are not represented by our model. Furthermore, allowing the model to simulate the buildup of perennial snow can lead to problematic behaviour due to the coarse grid cell resolution and the simplistic snow depletion approach (degree day approach) used in the model. In particular, the persistence of snow over multiple colder summers can result in the unrealistic buildup of a very large glaciated area that covers the entire grid cell. Given that our model is neither designed nor intended to simulate the evolution of glaciers, we prevent the simulation of this unrealistic process by removing any remaining snow with a summer peak at the beginning of August. We believe that this approach is a reasonable compromise that allows us to study permafrost dynamics excluding glacial dynamics.*

*We have added an explanation to clarify this model limitation:*

***While the assumption of a predominately seasonal snow cover is true for about 95\% of the Arctic permafrost region \citep{e.g.}{}{fontana2010perennial, trishchenko2018variations}, studies suggest that during the Little Ice Age the Canadian High Arctic Islands were more substantially characterized by perennial snowfields \citep{andrews1976little} and glaciers \citep{williams1978little}. It is important to note that this has implications for the interpretation of the model results, as the snow scheme used may not fully capture conditions in the very high latitudes and mountainous regions that may have been affected by year-round snowpacks, ice fields, or glaciers.***

L149 – Ground stratigraphy section 2.2 – Not much information is provided on bedrock stratigraphy, only soil stratigraphy – see earlier comments.

*We have added descriptions on how the bedrock layer is treated in the method section (please see response above) and also in the subsections 2.2.1 and 2.2.2.*

***Below $z_{\rm S}$, a bedrock layer extends down to the end of the vertical model domain.***

***The bedrock layer is represented by a purely mineral soil layer with a reduced water/ice content.***

L213 – What is used for the water/ice content of bedrock?

*A uniform water/ice content was assumed for the bedrock layer, which was drawn from a uniform distribution between 1% and the mineral soil porosity from the input dataset. This is indicated in Table 1 and now clearly stated in the method section (please see response to major comment above).*

L251-254 – See earlier comment regarding depth of temperatures included in Biskaborn et al. (2019). Have only the boreholes that have temperatures reported at 10 m depth been utilized in the model evaluation. This would reduce the amount of information available that could be utilized for model validation and also means that some regions are not represented.

*We incorporated all boreholes from Biskaborn et al. 2019 which are located within our model domain, irrespective of the depth of observations. As explained above, we used the model output at the depth closest to the observation depth for the model-observation comparisons. We have changed the figure caption accordingly (see response above) and added the following explanation here:*

***For the model-observation comparison, the model output is compared at the depth closest to the measurement depth.***

L256 – Deviation of up to 2K seems rather large. There is no real consideration of vegetation which is an important factor influencing the ground thermal regime. This could be a key factor responsible for the deviation.

*We agree that the neglection of vegetation is a limitation of our current model setup. However, a sophisticated treatment of vegetation and its effects on the surface temperature would require the extension of our model by a*

*surface energy balance (and, to be consistent, a hydrology scheme). This would not only require additional forcing data (e.g. radiation), which are less well constrained for the paleo period, but likely also increase the computational demand and with that the ability to perform large parameter ensemble simulations.*

*As the goal of our study was not to reproduce the thermal state of all individual boreholes, but rather to capture the overall thermal dynamics of the permafrost region over the past centuries, we consider the deviations for some boreholes acceptable, especially since we are not aware of other large-scale modeling studies which reported much-improved fits to borehole observations.*

*In the revised version we have added a detailed discussion of the uncertainties that arise from not explicitly accounting for vegetation in our simulations (see detailed response later).*

L265-269 – The other thing that may be important in mountainous terrain is that there may be little soil and organic material. This along with bedrock conditions means ground temperatures will closely track air temperature.

*For the mountainous terrain we found an RMSE of 3.1K and a warm bias of 1.65K in the simulations. From our experience, reducing the soil and organic material would even increase this bias, as these usually have a cooling effect. Therefore, we suspect that the deviation is rather due to the neglect of orographic effects in the climate forcing and potentially a bias in the representativeness of the borehole locations. In addition, an overestimation of the snow height in the simulations might be the case for exposed boreholes.*

Figure 2 – Some clarifications are required. Be clear what the reference period is for the anomaly calculation. If I understand correctly, the map (a) only shows the mean MAGT calculated over a decadal period although the way the caption is written it may imply the map shows trends. The graph in (c) compares observed and simulated trends in MAGT, I assume between 2007-16 and this should be clear in the caption in the description of (c).

*We agree and have modified the caption accordingly.*

**The map (a) shows modeled mean annual ground temperatures (MAGT) at 10 m depth averaged over the decade from 2007 to 2016. Circles indicate temperature deviations between the observed ground temperatures of the Global Terrestrial Network for Permafrost (GTN-P) boreholes and the modeled temperatures at the depth of the subsurface grid closest to the depth of the respective observations. The scatter plot (b) illustrates the agreement between observed and modeled MAGT (2007-2016 averages) with observations lying in the same model grid cell being grouped together. On the y-axis the dots show the mean of the parameter ensemble while the whiskers show the range between the 5th and 95th percentiles. Scatter plot (c) illustrates the agreement between observed and modeled (ensemble-mean) trends in MAGT, each derived from a linear least-squares regression over the 2007-2016 period. Observed trends are only included if there are 5 or more years of observations available. Horizontal error bars indicate the range of all observed trends belonging to the same model grid cell. Vertical error bars correspond to the 5th and 95th percentiles of the trends estimated by the parameter ensemble.**

Figure 3 – Similar clarifications are required as mentioned for Figure 2. Does the map show average values for ALT? What is the period over which averages and trends are determined?

*We agree and have modified the caption accordingly.*

**Map (a) shows the simulated maximum annual thaw depth (Active Layer Thickness, ALT) averaged over the decade from 2007 to 2016. Circles indicate deviations between the ALT measurements of the Circumpolar Active Layer Monitoring (CALM) program and the modeled ALTs. Scatter plot (b) compares CALM measurements and modeled ALTs (2007-2016 averages). On the y-axis the dots indicate the mean of the model ensemble with error bars indicating the range between the 5th and 95th percentiles. On the x-axis the dots indicate the mean observed ALT thickness averaging all sites located within the corresponding model grid cell and the error bars show the range of observations if there is more than one observation in the corresponding grid cell. Dashed lines indicate relative deviations of $\pm20\,\%$ and the gray square indicates ALTs $<2\,$m for which a comparison between measurements and simulations is most meaningful. Scatter plot (c) shows modeled and measured trends in ALT over the 2007-2016 period for all sites with observations available for 5 or more years. Vertical error bars correspond to the 5th and 95th percentiles of the simulated ALT trends, and horizontal error bars indicate the range of observed trends for grid cells with**

***more than one corresponding measurement site.***

L280- 282 – You might consider comparison of tundra, shrub dominated and forested sites as the response of the shallow ground thermal regime and therefore ALT will be influenced by vegetation conditions.

*We thank the reviewer for this suggestion, and we agree that the vegetation cover could be one key reason explaining the generally poorer performance of the model in terms of ALT in lower latitudes of the model domain. Therefore, we extended the discussion in this section by emphasizing the vegetation cover as a critical control for ALT. We note, however, that also other factors such as observational errors and biases, and the definition of the model diagnostic for the ground depth can add to the explanation of the differences between observations and model results. Therefore, we decided* not to indicate *the vegetation cover in the figures, as this might give the impression that the vegetation is the main or single explanation for the deviations. We have extended the discussion of possible explanations for the deviations between simulated and modeled ALTs:*

***Several factors have to be considered for explaining the deviations between measured and modeled ALTs. First, it is important to point out that ALT measurements with poke probes are susceptible to a low-bias at sites with large ALTs, because gravel or other compact soil material could be falsely misinterpreted as permafrost. Second, in southern areas, CALM sites are preferentially located at known permafrost sites such as peatlands, which are, however, not representative of the large-scale picture. Third, our model does not fully take into account the protective effect of tall and dense vegetation covers, which would lead to shallower ALTs, especially in the Boreal biome \citep[e.g,.][]{stuenzi2022thermohydrological}. Fourth, our model does not simulate thaw subsidence and soil compaction processes, which would reduce ALTs, especially in ice-rich regions \citep[e.g.][]{guenther2015observing}. Therefore, perfect agreement between modeled and observed ALTs on a circum-Arctic scale is not expected. Nevertheless, the simulation demonstrates the high sensitivity of the simulated ALT to local ground water and ground ice contents, which are subject to large uncertainties and strong spatial variability. While the parameter ensemble simulations can provide insights into the associated model uncertainties, the actual spatial variability of ground water and ice content remains an unresolved challenge.***

L285 – 291 – Note that at some CALM sites, probing is done on grids as large as 1 km$^2$ so there are average ALT values available over larger area. Since ALT for most CALM sites is determined through probing, the thaw depth that can be determined is limited to less than 2 m and there is some bias in the data set with respect to the subsurface materials as probing can not be done in coarser material.

*We thank the reviewer for pointing out these further details on the CALM sites. We did not pre-select the CALM sites based on their probing area, so those sites with 1km² grids would be included in our comparison. However, the variability within one 1° by 1° cell might even exceed the variability within such a large CALM grid, such that we prefer to use the error bars in panel b to represent the variability within one model grid cell in case it contains multiple CALM sites. We also appreciated the fact that probing measurements beyond 2 m ALT are problematic by indicating a gray square in Fig. 3 panel b, and we discuss the problems with coarse or rocky ground material in l. 291f. In fact, these limitations of the CALM dataset would introduce a bias towards shallower ALTs in the observations, which might explain some of the deviations between the model and the measurements (see reply to previous comment).*

L291-292 – In the southern fringes of the permafrost regions, permafrost is largely limited to organic terrain. As mentioned in earlier comment the permafrost that formed during colder periods during the Little Ice Age in these areas continues to persist due to the thermal properties of the peat.

*As discussed in the reply to the major comment above, our model currently has a limited capability to represent ecosystem-protected permafrost in organic-rich peatlands. Here, our point is that permafrost monitoring programs such as CALM have a tendency to monitor those locations where permafrost occurs, even if it is only sporadic in overall area coverage. Therefore, our model simulations might be closer to the reality in these southern regions than the* comparison between *modeled and observed ALTs suggests.*

L305-308 – See earlier comments – The beginning of the period considered in the analysis overlaps with the latter part of the Little Ice Age so that the colder permafrost temperatures in the 18[th]-19[th] century would be a legacy of this period.

*We thank the reviewer for pointing out this connection. We have gratefully included it in the following explanation:*

*It is also worth noting that the L18C period coincides with the latter part of the "Little Ice Age," a generally colder period starting in the 13th century that affected large areas of the Northern Hemisphere \citep{miller2012abrupt}. Thus, the significantly lower permafrost temperatures are also partly a legacy of this period of persistently cold climatic conditions which predate the L18C period under study.*

L322-325 – The results (including 1970-1990 warming) for northern Quebec do not appear to agree with observations. Observed permafrost temperatures in the eastern Canadian Arctic, including sites in northern Quebec and the high Arctic (e.g. Alert) show that both air temperatures and ground temperatures cooled into the 1980s - 1990s with most of the warming occurring post about 1995 (see for example, Allard et al. 1995; Smith et al. 2010). Reconstructions of ground surface temperature fromborehole records also show this later initiation of warming in the Canadian High Arctic (Taylor et al. 2006) and northern Quebec (Chouinard et al. 2007).

*We thank the reviewer for pointing us to further observational evidence and past temperature reconstructions to evaluate our model. Indeed, we agree that various reports indicating a cooling during the late 1980s and 1990s seemingly contradict our model results which show a markedly warmer ground during the 1950-2000 period compared to 1850-1900. However, this seemingly opposing evidence is relativized upon further inspections.*

*For this, we obtained the air and ground temperature observations from two sites in northern Quebec (Salluit and Quaqtaq; Allard et al., 2020), and compared them to the respective model data from the closest gridcell of the model domain. For both sites, the measured air temperatures are systematically warmer by about 1-2 K compared to the ERA-Interim forcing, which is, however, well explainable by subgrid variability due to orographic effects or the proximity of the measurements sites to the sea. The observed mean annual ground temperatures (MAGT) at 10 m depth during the 2000s and 2010s lie very well within the model range and even agree well with the model mean, particularly for Quaqtaq, indicating that the model ensemble captures site-level conditions despite the bias in the forcing data. For the 20 m depth, additional borehole observations are available for the late 1980s and early 1990s from Allard et al., 1995. The differences between the observed temperature levels around 1990 and two decades later agrees well with the modeling results for Quaqtaq, while the model does not show a clear cooling around 1990 at Salluit. However, the observation period of seven years is rather short to draw firm conclusions. Overall, the observations and model results agree fairly well, particularly for Quaqtaq.*

*Following this evaluation of borehole data in the Quebec region we have added the following paragraph considering the climate reconstruction provided by Chouinard et al. (2007):*

*For the Québec region observations and model results agree well for the L20C period (Fig. Appendix a,b). However, we note that in the model forcing data before 1980 are significantly colder than those after 1980. This temperature shift is attributed to the applied paleo-temperature forcing. In particular, the cooling trend reconstructed by \citet{chouinard2007recent} for the decades around 1950 is not clearly visible in our forcing data. Nevertheless, the modeled temperature anomalies (Fig. \ref{fig:maps-MAGT}) indicate that the L20C period is much warmer than the L19C period. This finding is consistent with the observations and supported by regional climate reconstruction \citet{chouinard2007recent} revealing a longer-term warming trend in northern Québec starting in the 18th century.*

*In order to address this important aspect in our revised article, we extend the discussion on the modeled temperature anomalies and particularly explain the signal found for northern Quebec. We think that it is very important to further constrain the remaining discrepancies and think that our simulations will benefit from using more recent Reanalysis and Paleo simulations as forcing data in the future.*

L331-333 – The way this part is written it implies this conclusion is based on observational evidence. It would be better to say that "Simulations suggest that during the last decades, permafrost warming has occurred…" (see previous comment that this conclusion isn't fully supported by observations).

*We thank the reviewer and have clarified the formulations in this paragraph accordingly.*

L340 – In warmer permafrost especially with higher moisture/ice content, ground temperature profiles indicate isothermal conditions exist (e.g. Romanovsky et al. 2010; Smith et al. 2010) due to the phase change that is occurring. This should be mentioned to help explain this sharp change in ALT.

*This is indeed a very good point, we have included it per the suggestion of the reviewer.*

*In warmer permafrost, especially in regions with higher ground ice content, the temperature profile is isothermal and the permafrost is close to freezing point down to deep ground layers \citep[e.g.][]{romanovsky2010permafrost,smith2010thermal}. A small increase in MAGT under such conditions is sufficient to thaw deep soil layers, which explains the sharp increase in ALT.*

L349 – Section 3.4 (see also earlier comments) – The zonation of permafrost such as that presented on Brown et al. (1998) map, is not based on the depth of the permafrost table being less than a critical depth (e.g. 3m) which is essentially is being used in the analysis presented here. It is not really correct to say that a grid cell contains permafrost if the ALT is < 3m (L352-353). In bedrock ALT can be >3 m (see for e.g. figure 7 in Smith et al. 2010 and also Christiansen et al. 2010) and permafrost is still present.

*We agree with the reviewer, that our use of permafrost zonation is not consistent with the map of Brown et al. and also deviates from the criteria used by Obu et al. 2019. As explained in the response to the major comment above, we now avoid the terms continuous, discontinuous, etc., and instead only refer to permafrost occurrence probabilities. However, for the visualization of the permafrost extents, we chose the same thresholds (e.g. 0.5 to 0.9 for discontinuous permafrost) as Brown et al. and Obu et al. We further note that we included an additional diagnostic taking into account deeper ground layers, allowing for a better comparison with earlier assessments. The formulations regarding the loss of permafrost area have been changed accordingly in the revised manuscript.*

L386 – What is meant by "active thaw height"?

*We apologize for the inadvertently incorrect wording. Wording changed to* **"thaw depth"**

L391-392 – From the information presented, excess ice does not appear to have been considered.

*Correct, please see the following comment.*

L405-409 – Excess ice does not appear to be considered, only pore ice. There are also other factors related to surface water to consider. Changes in surface water distribution including lake drainage or shifting of rivers may also lead to permafrost formation. In coastal regions that have been undergoing and continue to undergo post-glacial uplift, permafrost is also forming.

*Yes, excess ice is not being considered in this study and we clarified this point in the revised version of the manuscript. We agree that changes in surface water distribution are affecting permafrost as well and that these changes are currently not captured by our model. We now mention the potential changes due surface water distribution and post-glacial uplift in the last point of the limitations:*

**In addition, the increase in permafrost areas due to the retreat of surface ice (glaciers and ice sheets), changes in surface water distribution (lakes and rivers), or post-glacial uplift in coastal regions is not taken into account.**

L410-415 – This lack of consideration of vegetation conditions is probably one of the most important limitations of the model both for simulations of current permafrost conditions and also the evolution of permafrost over longer periods. Just as snow is an important factor influencing the ground surface temperature, so is vegetation cover including forests, shrubs and mosses.

*We appreciate the reviewer's comments regarding the impact of vegetation on the subsurface thermal regime, and we agree that our current model may not fully capture this effect. However, our model setup does take into account the insulating layer at the surface, which is influenced by a wide range of factors such as water saturation and snow depth. While this may not fully represent the complex dynamics of a dynamically growing and responding vegetation layer, we believe that it provides a reasonable estimate of the temperature range for a variety of insulating surface conditions. At the same time, we acknowledge that a lack of vegetation is a limitation of the model, particularly in its ability to reproduce ecosystem-protected permafrost. As the reviewer notes, this means that our model may underestimate the extent of permafrost and its occurrence probability in particular at lower latitudes. Therefore, we will ensure that this limitation is clearly pointed out in the revised version of the manuscript, and state that future refinements of our model should account for the impact of vegetation on the subsurface thermal regime. The revised version now includes this paragraph on model limitations:*

*The lack of representation of vegetation limits the model's ability to accurately simulate current permafrost conditions and predict their long-term changes. Vegetation, like snow, plays an important role in regulating the surface temperature of the ground through its composition and structure. Vegetation acts as an insulating layer and affects the amount of solar radiation reaching the ground, as well as heat and moisture exchange with the atmosphere \citep[e.g.][]{beringer2001presentation,stuenzi2022thermohydrological,heijmans2022tundra}. Neglecting the influence of vegetation leads to an underestimation of the extent of permafrost regions and their expansion to lower latitudes, where our current model tends to overestimate ALT values. Although vegetation is not explicitly represented, it is important to note that our model accounts for a range of isolating surface conditions. In this way, the parameter ensemble simulations performed partially emulate thermal buffer effects, such as those caused by organic surface layers, and provide an estimate of the potential probability of permafrost occurrence.*

L431-432 – Vegetation may also be a key factor here as well.

*We agree with the reviewer and added this factor as a major limitation. Please see the response above.*

L440-443 – See earlier comments about basing permafrost occurrence on depth of permafrost table.

*We have addressed this issue in the revised version of the manuscript - please see earlier replies.*

References

Allard M, Wang B, Pilon JA (1995) Recent cooling along the southern shore of Hudson Strait Quebec, Canada, documented from permafrost temperature measurements. Arctic and Alpine Research 27:157-166

Chouinard C, Fortier R, Mareschal JC (2007) Recent climate variations in the subarctic inferred from three borehole temperature profiles in northern Quebec, Canada. Earth and Planetary Science Letters 263:355-369

Christiansen HH, Etzelmuller B, Isaken K, Juliussen H, Farbot H, Humlum O, Johansson M, Ingeman-Neilsen T, Kristensen L, Hjort J, Holmlund P, Sannel ABK, Sigsgaard C, Akerman J, Foged N, Blikra LH, Pernosky MA, Odegard R (2010) Thermal state of permafrost in the Nordic area during the IPY 2007-2009. Permafrost and Periglacial Processes 21:156-181

Froese DG, Westgate JA, Reyes AV, Enkin RJ, Preece SJ (2008) Ancient permafrost and a future, warmer Arctic. Science 321:1648

Holloway JE, Lewkowicz AG (2020) Half a century of discontinuous permafrost persistence and degradation in western Canada. Permafrost and Periglacial Processes 31:85-96. doi:10.1002/ppp.2017

James M, Lewkowicz AG, Smith SL, Miceli CM (2013) Multi-decadal degradation and persistence of permafrost in the Alaska Highway corridor, northwest Canada. Environmental Research Letters 8 045013:10. doi:10.1088/1748-9326/8/4/045013

Lachenbruch AH, Marshall BV (1986) Changing climate: geothermal evidence from permafrost in the Alaskan Arctic. In:  Science, vol v. 234. pp p.689- 696

Romanovsky V, Isaksen K, Drozdov D, Anisimov O, Instanes A, Leibman M, McGuire AD, Shiklomanov N, Romanovsky VE, Smith SL, Christiansen HH (2010) Permafrost thermal state in the polar Northern Hemisphere during the International Polar Year 2007-2009: a synthesis. Permafrost and Periglacial Processes 21:106-116

Smith SL, Walker D (2017) Chapter 4, Changing permafrost and its impacts. In:  Snow, Water, Ice and Permafrost in the Arctic (SWIPA) 2017. Arctic Monitoring and Assessment Program (AMAP) Oslo, Norway, pp 65-102

Smith SL, Romanovsky VE, Isaksen K, Nyland KE, Kholodov AL, Shiklomanov NI, Streletskiy DA, Drozdov DS, Malkova GV, Christiansen HH (2022) [Arctic] Permafrost [in "State of the Climate in 2021"]. Bulletin of the American Meteorological Society 103 (8):S286-S290. doi:10.1175/BAMS-D-22-0082.1

Smith SL, Romanovsky VE, Lewkowicz AG, Burn CR, Allard M, Clow GD, Yoshikawa K, Throop J (2010) Thermal state of permafrost in North America - A contribution to the International Polar Year. Permafrost and Periglacial Processes 21:117-135. doi:10.1002/ppp.690

Treat CC, Jones MC (2018) Near-surface permafrost aggradation in Northern Hemisphere peatlands shows regional and global trends during the past 6000 years. The Holocene 28 (6):1000-1010. doi:10.1177/0959683617752858

Williams PJ, Smith MW (1989) The Frozen Earth: fundamentals of geocryology. Cambridge University Press, Cambridge, U.K.

**Citation**: https://doi.org/10.5194/egusphere-2022-473-RC1

---

## Author Comment (AC3)

**Reply letter to reviewer 2**

*We thank the reviewer for the thorough review and valuable suggestions for improvement. We have addressed all points in the following. All responses are in italics and changes made in the manuscript are highlighted in bold. Updated figures are shown, but may not be exactly as shown later in the revised version due to ongoing graphics editing.*

General Comments

The article describes a new permafrost model, labelled CryoGridLitle, focused on minimizing the computational and forcing resources required to be executed while achieving a realistic representation of permafrost areas. Furthermore, the analysis presents a comprehensive estimate of the uncertainties associated to the model, with the only exception of the uncertainties due to the external forcing and the model itself. The paper is clearly written and structured but I have concerns about the suitability of the model evaluation process followed by the authors, as it is limited in scope. I think that the topic is adequate for the journal, and that the development of a permafrost model able to perform multi-century simulations is relevant for the climate and cryosphere communities. I recommend publication after major revisions.

*We thank the reviewer for his thorough review of our manuscript and note the concerns about the limited evaluation of the model. In response, we have made substantial improvements to the manuscript and have included additional material focusing on both the assessment of the model physics and a more detailed assessment at key sites representing different permafrost conditions and histories, as suggested by the reviewer.*

Major points

M1- The authors cite in the Introduction a series of works in which permafrost models are used in paleoclimate studies. Nevertheless, the performance of the CryoGridLitle is not compared to that from other models applied at paleoclimate scales. Is the CryoGridLitle framework performing better? What are the advantages of this new model in comparison with previous models used in paleoclimate studies? What are the disadvantages? The answer to these questions should appear in an article introducing a new modelling framework when other models are already available.

*We agree with the reviewer that the advantages, limitations, and goals of our modeling approach should be stated more clearly in the introduction. In the revised version of the manuscript, we have rewritten and expanded the relevant paragraph as follows:*

**Here, we present and evaluate a computationally efficient numerical permafrost model (CryoGridLite) designed to provide insights into the evolution of the thermal state of permafrost and active layer thickness over many centuries for the Arctic permafrost region. With the CryoGridLite model, we aim to bridge the gap between very sophisticated permafrost process models and reduced schemes used for paleoclimatic simulations. Compared to comprehensive process models like CryoGrid3 \citep{westermann2016} or the CryoGrid Community Model \citep{westermann2023cryogrid}, CryoGridLite is approximately three orders of magnitude faster, enabling the execution of long-term simulations spanning hundreds to thousands of years. The enhanced efficiency is achieved through two key components: (i) the utilization of an implicit solution scheme and (ii) a streamlined representation of the underlying processes.**
**Our approach aligns with the objective of providing plausible ranges of permafrost states rather than focusing on highly precise results for specific locations. This is accomplished through the application of multi-parameter ensemble simulations, which have become feasible due to the substantial enhancements achieved in computational efficiency. By employing this methodology, we can capture a broader range of potential permafrost conditions, encompassing the inherent uncertainties and variability associated with real-world scenarios. In the simulations we account for uncertainties in soil parameters such as water and ice content, and uncertainties in snowpack properties.**

M2- The ERA Interim reanalysis and the MK3Lv.1.2 model are now outdated. ERA5 and ERA5-Land are state-of-the-art reanalyses with better spatial resolution. And the same can be said about paleosimulations included in the PMIP4/CMIP6 projects. Why were these old products selected as forcings of the simulations? As the authors indicate in the manuscript, the spatial resolution of the forcings is one of the aspects affecting the performance of the CryoGridLite model, and those forcings need to be interpolated before running the model. Would not make more sense to use new, high resolution renalyses and paleosimulations for this study?

*We appreciate the reviewer's comment regarding the use of older forcing data in our study. While we acknowledge that more recent reanalysis and paleo simulations exist, our study was based on ERA Interim and the MK3Lv.1.2 forcing data due to their availability within our data framework and their widespread acceptance when we started developing CryoGridLite. We understand that it is desirable to use the most recent forcing data, but we would like to stress that new forcing data do not necessarily invalidate the applicability of older data. For instance, potential biases have been reported for ERA5-Land data in the Arctic (Cao et al., 2020), while applicability of ERA Interim has been validated in numerous studies.*

*Our study aims to demonstrate the capabilities of our modeling approach to perform long-term permafrost simulations that also consider highly uncertain and variable soil and snow parameters using large ensembles. We agree that future studies focusing on investigating the response of permafrost within specific climate periods or related to specific climatic events in the past should use state-of-the-art forcing data. However, for this study, which focuses on model applicability and validity, we think that the use of older forcing data is acceptable. Preprocessing and repeating all ensemble simulations would be a very excessive task and using a different forcing will not change the broad scale thermal history. Nonetheless, we appreciate the reviewer's critique, and we will explicitly mention in the revised manuscript that more recent forcing data should be used in follow-up studies focusing on model application rather than general model validation.*

*Cao, B., Gruber, S., Zheng, D., and Li, X.: The ERA5-Land soil temperature bias in permafrost regions, The Cryosphere, 14, 2581–2595, https://doi.org/10.5194/tc-14-2581-2020, 2020.*

M3- After reading the two first sections, it is clear that the main advantage of the CryoGridLitle is the speed of execution in comparison with more comprehensive permafrost medelling schemes. To this end, many important processes are not included in the model, such as water advection or runoff. Nevertheless, there is no estimate about the performance gain of CryoGridLitle in comparison with other schemes. Comparing the time required to simulate the 1400 years of this study in the CryiGridLitle and in another scheme (maybe CryoGrid3?) would allow the reader to assess if the gains in execution time really compensate for the absent processes.

*We agree with the reviewer. We performed a runtime comparison and added it to the relevant paragraph (see response above). In addition, we have performed a thorough performance test of the implicit solution scheme and added the following description to the method section, including an additional figure:*

***The numerical performance and stability of the applied implicit method is evaluated in comparison with commonly used numerical integration methods: Crank-Nicolson and TRBDF2 (2nd order diagonally implicit Runge-Kutta methods), Radau IIA5 (5th order fully implicit tableau method), Heun (canonical explicit trapezoidal method), and SSPRK43 (4th order stability-preserving explicit Runge-Kutta method). The test simulations under periodic freeze-thaw conditions result in a mean absolute error of about \qty{0.014}{K} for time steps of 24 hours. With this acceptable level of accuracy, the CryoGridLite scheme is about an order of magnitude faster than the other numerical integration schemes.***

[Figure]

*Fig. xyz: Work-precision plot comparing different numerical integration methods with CryoGridLite on heat conduction with phase change with periodic upper boundary. The y-axis shows the computational demand in CPU time of each numerical solve while the x-axis shows the global temperature error of solution with respect to a high-resolution reference simulation using a 10-stage, 4th order strong stability preserving (SSP) Runge-Kutta with a 60 second timestep. All of the algorithms (except CryoGridLite) use adaptive time stepping schemes with tolerances changed from $10^{-6}$ to $10^{-1}$. Since CryoGridLite is a fixed-time step method, we instead use a timestep of 24 hours scaled from $10^{-3}$ to $10^{0}$ for comparison. With the largest timestep of 24 hours CryoGridLite achieves an acceptable global error of approximately 0.014 K while being an order of magnitude more efficient than all other methods.*

M4- The authors should consider the possibility of adding some theoretical tests to evaluate the ability of the model to simulate physical processes, such as propagation of heat with different thawing conditions. I am aware that the CryoGrid3 model is able to successfully reproduce such processes, but since the formulation for heat propagation in the CryoGridLitle model has been modified, there should be some proof that the model is able to correctly reproduce phase change and basic heat diffusion though the ground.

*We thank the reviewer for this recommendation, which we have included in the revised version of the manuscript. In the appendix we will present a theoretical test against an analytical solution for heat transfer without phase change that shows the ability of the model to correctly reproduce heat diffusion. We will also show a test against the Stefan solution that demonstrates the ability of the model to correctly simulate the progression of the thawing front.*

[Figure]

**Fig. xyz: Evaluation of CryoGridLite in simulating heat diffusion with sinusoidal upper and zero flux lower boundary conditions without phase change. The analytical solution (a) for a semi-infinite half-space is compared to the solution obtained using CryoGridLite (b). Analysis of the differences (c) reveals a relatively small periodic bias of less than ±0.01K at deeper depths due to the spatial discretization. These results demonstrate that the numerical scheme effectively captures the heat diffusion process, yielding results that closely align with the analytical solution.**

[Figure]

**Fig. xyz: Comparison of the analytical solution to the two-phase Steefan problem and CryoGridLite for the one-sided thawing of a frozen soil column. The soil temperature is initialized at -0.02°C, and a constant upper boundary temperature of 1°C is applied for a period of 5 years. The spatial domain is discretized into a nonuniform rectangular grid cell with minimum spacing of 0.01 m down to a depth of 0.5 m. The good agreement between the analytical solution and the CryoGridLite simulation demonstrates that the model is able to accurately represent phase change of water.**

M5- Another concern is the evaluation of the model against measurements of soil temperature and active layer thickness (ALT) at CALM sites. I wonder if evaluating soil temperatures and ALT from CALM is enough to "validate" the model. The CALM network is indeed a very valuable resource for model assessment, but it is also limited geographically and in time. Otherwise, there are several estimates of near-surface permafrost area in the present, some of the relevant articles have been cited in the manuscript, and several indices have been developed in order to use meteorological data as a benchmark for evaluating permafrost models (e.g., 1-2). Then, why is the evaluation of CryoGridLite restricted to comparing with CALM stations? At the very least, a comparison with estimates of permafrost area during recent times should appear in the article, as this is a relevant factor to evaluate the quality of the simulations.

*We appreciate the reviewer's comment and would like to clarify that while we are validating the model for Active Layer Thickness (ALT), we are also validating it for thermal state representation using borehole temperature measurements. Unlike other modeling studies, we are not limiting our validation to temperature alone but also testing the reproducibility of trends in the soil thermal regime. However, we agree that further validation can be done to improve the model's robustness. As suggested by both reviewers we have collected additional data, especially from long-term borehole temperature measurements, to improve our model's validation. Furthermore, we will also validate the model's ability to reproduce recent permafrost extent as suggested by the reviewer.*

***A more detailed model evaluation is being conducted using borehole measurements, where long-term data spanning over 10 years are available for selected sites. Air and ground temperature observations from four sites in Canada (Salluit, Quaqtaq, and Ellesmere Island) \citep{allard2020borehole} and Russia (Urengoy) \citep{smith2022} are being obtained and compared to the corresponding model data from the nearest grid cell within the model domain. For the sites in the Quebec region (Salluit and Quaqtaq), the measured air***

temperatures systematically show a difference of about \qtyrange{1}{2}{K} compared to the ERA interim forcing. This can be well explained by small scale variability e.g. due to orographic effects or proximity of the measurement sites to the coast. For both sites, the observed mean annual ground temperatures (MAGT) at \qty{10}{m} depth during the 2000s and 2010s are very well within the modeled ensemble range and even agree well with the model mean, especially for Quaqtaq, suggesting that the model ensemble captures site-level conditions despite a bias in the forcing data. For a depth at \qty{20}{m}, additional borehole observations are available for the late 1980s and early 1990s \citep{allard1995recent}. The time series of measurements, however, is not long enough to confirm the pronounced negative temperature anomaly simulated before 1980 for Quaqtaq. For a high Arctic site on Ellesmere Island (Fig. Appendix c), the long-term observations and simulations also appear to agree reasonably well, although the natural variability is not fully captured by the model. The long-term measurements from Urengoy (Fig. Appendix d) in Russia show very cold borehole temperatures, and only the coldest simulations in the ensemble show such low temperatures. Another borehole is located in the immediate vicinity which shows much warmer temperatures but has a shorter time series.

[Figure]

*Figure Axyz: Model evaluation based on long-term borehole temperature and near-surface air temperature measurements in different regions. (a) Quaqtaq, (b) Salluit in the Quebec region, (c) a high Arctic region on Ellesmere Island, and (d) Urengoy in northern West Siberia, Russia. If multiple borehole measurements are available, they are indicated separately.*

M6- Lines 309-319: There are some works providing ground surface temperature histories from inversions of deep subsurface temperature profiles in the North American part of the Arctic (e.g., 3-6). Nevertheless, the authors choose proxy-based paleoreconstructions of surface air temperature to compare with permafrost temperature changes in Alaska. Inversions of subsurface temperature profiles are more adequate for this comparison, since these are estimates of changes in ground surface temperature.

*We thank the reviewer for this comment and have gratefully incorporated the paleoreconstructions of surface temperatures into the discussion of the revised version:*

*For the Québec region observations and model results agree well for the L20C period (Fig. Appendix a,b). However, we note that in the model forcing data before 1980 are significantly colder than those after 1980. This temperature shift is attributed to the applied paleo-temperature forcing. In particular, the cooling trend reconstructed by \citet{chouinard2007recent} for the decades around 1950 is not clearly visible in our forcing*

*data. Nevertheless, the modeled temperature anomalies (Fig. \ref{fig:maps-MAGT}) indicate that the L20C period is much warmer than the L19C period. This finding is consistent with the observations and supported by regional climate reconstruction \citet{chouinard2007recent} revealing a longer-term warming trend in northern Québec starting in the 18th century.*

Minor points

m1- The title should be changed to reflect the model evaluation part of the study. Something like "The evolution of Arctic permafrost over the last three centuries using a new, fast permafrost model".

We appreciate the suggestion and modified the title which now reads: "**The evolution of Arctic permafrost over the last three centuries from ensemble simulations with the CryoGridLite permafrost model**"

m2- Lines 103-106: It is not clear how the model can produce deviations  larger than the threshold and still conserve energy. Could you please give more detail about this point?

*The model employs the energy form to solve the heat equation, from which the temperature profile is derived. The heat flux within the soil profile is determined using a tridiagonal solver that calculates the redistribution of heat by providing the gradients based on which the energy is distributed. This iterative procedure continues until the solved temperature profile reaches equilibrium with the temperature profile diagnosed from the energy profile. However, due to the discontinuity caused by the freezing curve of water, there may be discrepancies between the temperature profile and the energy profile. In most integration steps, this temperature discrepancy is below the selected threshold (<0.01 K), but in some cases, satisfactory conversion cannot be achieved within an acceptable number of iterations (500). This only affects three grid cells around which the actual freezing front is located and occurs only if a cell is almost completely thawed/frozen and the thawing/freezing front is transferred to the neighboring cell. This discrepancy only indicates the energy is not exactly distributed as expected from the solution of the temperature profile. The total energy content resulting from the fluxes across the boundaries and the previous heat content remains conserved for each time step.*

m3- The definition of 1850-1900 as preindustrial period is not consistent with the consensus in paleoclimate publications. Indeed, the period 1750-1800 is a better option for preindustrial times (3). Please, consider a change of labelling, perhaps with 1850-1900 as later 19th century (L19C).

*We thank the reviewer for this remark and have changed the labeling accordingly throughout the manuscript.*

m4- I am unable to see the mentioned short-term changes in permafrost area due to volcanic eruptions in Figure 6. Is there any other way of presenting the results of Figure 6 that shows more clearly the effect of the eruptions?

*We agree with the reviewer. The effects of volcanic eruptions are not well visible in the figure. Also, in response to a comment from reviewer 1 and a suggestion from the initial editorial review, we have created a new diagnostic to evaluate the presence of near-surface permafrost. This now clearly shows the effects of volcanic eruptions on permafrost occurrence probability.*

[Figure]

[Figure]

**Figure 6. Total areas of probability of near-surface permafrost occurrence according to two different diagnostics focusing on (a) 3 m and (b) 10 m ground depth. The zones of occurrence probability are derived from the parameter ensemble simulations. Horizontal lines mark areas of permafrost regions as delineated by Brown et al. (1997) and Obu et al. (2019) for the early 21st century. Dashed red vertical lines mark strong volcanic eruptions events (Volcanic Eruption Index (VEI) ≤ 6) which are represented in applied climate forcing data.**

References

1- Koven, C. D., Riley, W. J., and Stern, A.: Analysis of Permafrost Thermal Dynamics and Response to Climate Change in the CMIP5 Earth System Models, Journal of Climate, 26, 1877–1900, https://doi.org/10.1175/JCLI-D-12-00228.1, 2012/10/01.

2- Burke, E. J., Zhang, Y., and Krinner, G.: Evaluating permafrost physics in the Coupled Model Intercomparison Project 6 (CMIP6) models and their sensitivity to climate change, The Cryosphere, 14, 3155–3174, https://doi.org/10.5194/tc-14-3155-2020, 2020.

3- Majorowicz, J. A., Skinner, W. R., and Šafanda, J.: Large ground warming in the Canadian Arctic inferred from inversions of temperature logs, Earth and Planetary Science Letters, 221, 15–25, https://doi.org/https://doi.org/10.1016/S0012-821X(04)00106-2, 2004.

4- Taylor, A. E., Wang, K., Smith, S. L., Burgess, M. M., and Judge, A. S.: Canadian Arctic Permafrost Observatories: Detecting contemporary climate change through inversion of subsurface temperature time series, Journal of Geophysical Research: Solid Earth, 111, https://doi.org/https://doi.org/10.1029/2004JB003208, 2006.

5- Taylor, A. E. and Wang, K.: Geothermal inversion of Canadian Arctic ground temperatures and effect of permafrost aggradation at emergent shorelines, Geochemistry, Geophysics, Geosystems, 9, https://doi.org/https://doi.org/10.1029/2008GC002064, 2008.

6- Jaume-Santero, F., Pickler, C., Beltrami, H., and Mareschal, J.-C.: North American regional climate reconstruction from ground surface temperature histories, Climate of the Past, 12, 2181–2194, https://doi.org/10.5194/cp-12-2181-2016, 2016.

7- Hawkins, E., Ortega, P., Suckling, E., Schurer, A., Hegerl, G., Jones, P., Joshi, M., Osborn, T. J., Masson-Delmotte, V., Mignot, J., Thorne, P., and van Oldenborgh, G. J.: Estimating Changes in Global Temperature since the Preindustrial Period, Bulletin of the American Meteorological Society, 98, 1841–1856, https://doi.org/10.1175/BAMS-D-16-0007.1, 2017.

**Citation**: https://doi.org/10.5194/egusphere-2022-473-RC2